# An interhemispheric neural circuit allowing binocular integration in the optic tectum

Christoph Gebhardt [1], Thomas O. Auer [1], Pedro M. Henriques[2], Gokul Rajan [1,3], Karine Duroure[1,3], Isaac H. Bianco [2,4]* & Filippo Del Bene [1,3,4]*

Binocular stereopsis requires the convergence of visual information from corresponding points in visual space seen by two different lines of sight. This may be achieved by super-position of retinal input from each eye onto the same downstream neurons via ipsi- and contralaterally projecting optic nerve fibers. Zebrafish larvae can perceive binocular cues during prey hunting but have exclusively contralateral retinotectal projections. Here we report brain activity in the tectal neuropil ipsilateral to the visually stimulated eye, despite the absence of ipsilateral retinotectal projections. This activity colocalizes with arbors of commissural neurons, termed intertectal neurons (ITNs), that connect the tectal hemispheres. ITNs are GABAergic, establish tectal synapses bilaterally and respond to small moving stimuli. ITN-ablation impairs capture swim initiation when prey is positioned in the binocular strike zone. We propose an intertectal circuit that controls execution of the prey-capture motor program following binocular localization of prey, without requiring ipsilateral retinotectal projections.

[1] Institut Curie, PSL Research University, INSERM U934, CNRS UMR3215, Paris, France. [2] Department of Neuroscience, Physiology & Pharmacology, University College London, London WC1E 6BT, UK. [3] Present address: Sorbonne Université, INSERM, CNRS, Institut de la Vision, Paris, France. [4] These authors jointly supervised: Isaac H. Bianco, Filippo Del Bene.  *email: i.bianco@ucl.ac.uk; filippo.del-bene@inserm.fr

In order to feed on moving prey, predatory animals must recognize, pursue, and finally capture their target. Hunting sequences of zebrafish larvae consist of a pursuit phase that sometimes culminates in a capture maneuver[1,2]. The pursuit phase is characterized by discrete approach swims (AS) and asymmetric bouts (J-turns and high-angle turns) that enable the larva to orient toward and approach its prey[1,3–6]. Capture maneuvers consist of specific high-acceleration capture swims (also called rams) or suction feedings[1,7].

Having two forward facing eyes with a binocular field of view may enable predating animals to evaluate target distance (binocular stereopsis) and thereby increase the likelihood of successfully capturing prey. Zebrafish larvae have laterally positioned eyes resulting in a negligible binocular field of view under most behavioral conditions. However, during hunting sequences, larvae consistently converge their eyes, resulting in a substantial increase in their binocular visual field[3]. Vergence angle varies with target distance[8] and capture swims are initiated once the target is located in a stereotypic strike zone within the binocular visual field, around 500 μm directly in front of a larva[1,3,8]. Furthermore, prey capture is strongly impaired when retinal input is unilaterally removed[9]. This evidence suggests that zebrafish larvae use binocular depth cues to locate and capture prey.

A binocular stereopsis mechanism for localizing an object in a three-dimensional environment depends on visual information about corresponding object features, as viewed by each eye, converging on binocular downstream neurons. For example, in mammals, it has been suggested that this convergence is achieved by direct superposition of corresponding retinal input via ipsi- and contralaterally projecting optic nerve fibers from the two eyes[10]. However, in zebrafish larvae, retinotectal fibers project exclusively contralaterally[11]. It is thus unclear how a convergence of binocular visual information for localizing prey in the strike zone is accomplished.

The optic tectum (OT) is the main retinorecipient area in the midbrain of teleosts and is homologous to the superior colliculus (SC) in mammals. It is an important sensorimotor integrator that is essential for visually guided behaviors including predator avoidance[12] and prey capture[9,13]. The OT consists of two main areas: a synaptic neuropil and a cellular *stratum periventriculare* (SPV). The OT neuropil receives input from contralateral retinal ganglion cells (RGCs)[11] which innervate four retinorecipient layers, from superficial to deep: the *stratum opticum* (SO), the *stratum fibrosum et griseum superficiale* (SFGS), the *stratum griseum centrale* (SGC), and the *stratum album centrale* (SAC)[14,15].

In this study, we report the surprising discovery of activity in the deep layers of the tectal neuropil in response to visual stimulation of the ipsilateral eye of zebrafish larvae. We identify a population of intertectal interneurons, ITNs, that responds to visual motion and innervates both left and right OT. ITNs are GABAergic and establish synapses in the deep layers of the neuropil. Furthermore, these neurons respond to small moving spots that resemble prey of zebrafish larvae such as *Paramecia*. ITN-ablation uncovers a specific requirement of these neurons for the initiation of capture swims when prey is positioned in the binocular strike zone directly in front of the larva. We propose that ITNs are a central component of a bilateral neural circuit that integrates binocular visual information to enable localization and capture of prey, in the absence of ipsilateral retinotectal projections (IRPs).

## Results

### Visually evoked activity in the ipsilateral tectum. First, we wanted to know if visual information from one eye is represented in the ipsilateral tectal hemisphere, despite the lack of ipsilaterally projecting retinotectal axons. To this end, we used 2-photon microscopy to image the OT of 5 days-post-fertilization (dpf) larvae that expressed the calcium indicator GCaMP5G under control of a pan-neuronal promoter. One eye was visually stimulated with moving bars (width: 9°, speed: 20° s$^{-1}$, 12 angular directions spaced 30° apart) and the second eye was surgically removed prior to functional imaging to ensure monocular visual input (Fig. 1a). The absence of regenerative re-routing of retinal fibers from the remaining eye to the non-innervated tectal hemisphere after monocular enucleation was confirmed by anterograde DiO tracing (Supplementary Fig. 1). After correcting for motion artefacts, image timeseries from each larva were registered to the corresponding *z*-plane of a 5 dpf *Tg(elavl3: GCaMP5G)* reference larva. We computed average stimulus-triggered fluorescence responses for 6 anatomical subregions of the OT (SPV, deep and superficial neuropil in the ipsi- or contralateral tectal hemisphere, Fig. 1a). Consistent with fully crossed retinotectal projections in zebrafish, we observed strong calcium transients in all 3 regions of the tectum contralateral to the stimulated eye (Fig. 1b, upper three traces). Surprisingly, we also detected visually evoked responses in the tectal neuropil ipsilateral to the stimulated eye, i.e., contralateral to the enucleated eye (Fig. 1b, lowest two traces). This activity appeared confined to the neuropil as almost no response to moving bars was detected in ipsilateral periventricular neurons (PVNs) (Fig. 1b, ipsilateral SPV trace). To generate anatomical maps of stimulus-correlated activity, we used a regression-based approach[16] and, consistent with our previous observations, found highly-correlated voxels in the tectal neuropil and SPV contralateral to the stimulated eye but also within the deep laminae of the ipsilateral tectal neuropil (comprising SGC and SAC) (Fig. 1c, d).

### ITNs connect the two hemispheres of the optic tectum. To explain these observations, we hypothesized that a population of commissural neurons might exist that preferentially targets the deep laminae of the tectal neuropil. Moreover, this population would likely be visual motion responsive and directly or indirectly activated by retinal afferents.

Our lab isolated a *Gal4*-expressing transgenic zebrafish line, *Gal4ic3034Tg* (referred to as $^{ITN}Gal4$ hereafter), which labels a population of neurons interconnecting the two tectal hemispheres (Fig. 2a, b). Based on this anatomical feature we named the neuronal population intertectal neurons (ITNs). In addition to labeling ITNs, the *Gal4ic3034Tg* transgene is also expressed in the pineal gland, spinal cord, superficial interneurons in the OT (SINs), and sparsely in PVNs (Fig. 2a). The number of ITNs labeled in $^{ITN}Gal4$, *UAS:GFP* larvae varied across animals with a minimum of 22 labeled cells (12 on the right, 10 on the left) and a maximum of 48 (27 right, 21 left) (average from 18 randomly chosen larvae: 32 ITNs in total, 16 per side). This variability is likely due to variegation in transgene expression[17].

The cell bodies of ITNs are situated in two bilaterally symmetric nuclei below the OT in the mesencephalic tegmentum (Fig. 2a, lower left panel), and send their neurites dorsally, crossing the tectal commissures in ladder-like trajectories (Fig. 2a, upper left panel). ITN neurites develop concomitantly with retinal innervation of the OT: They cross the midline and arborize in the OT starting from 3.5 dpf as revealed by single ITN neurite tracing (Fig. 2c). By 4 dpf, individual ITNs form increasingly complex arborization patterns in both the ipsi- and contralateral OT neuropil (Fig. 2d). Arbors of individual ITNs localize to the deep neuropil layers, between the SGC and the SAC (Supplementary Fig. 2a, b).

2-photon imaging of ITN somata revealed that cells contralateral to the stimulated eye showed visually evoked activity in response to moving bars (Supplementary Fig. 2c, d) indicating

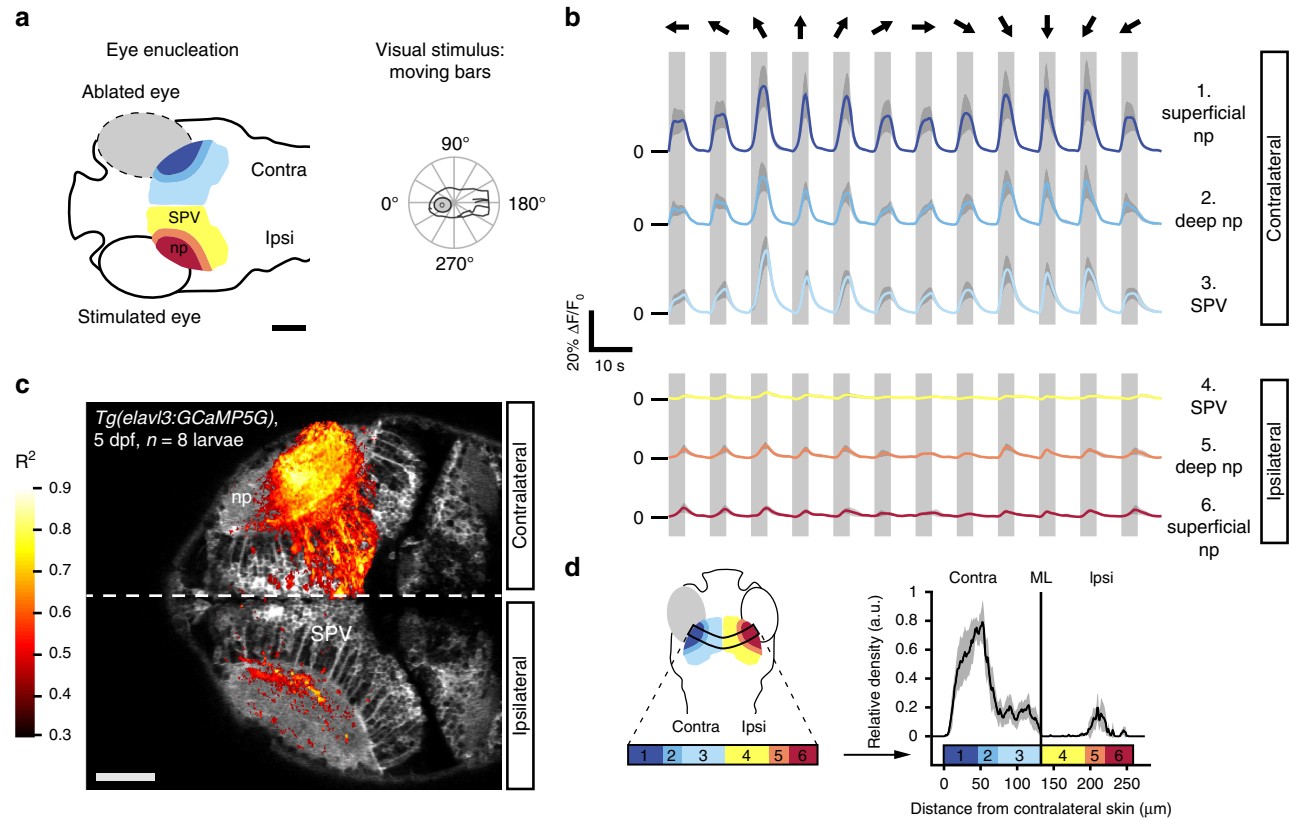

**Fig. 1** Visually evoked activity in the deep neuropil of the tectal hemisphere ipsilateral to the visually stimulated eye. **a** Optic tecta of 5 days-post-fertilization (dpf) larvae expressing GCaMP5G were imaged after monocular enucleation at 3–4 dpf. The remaining eye was visually stimulated with moving bars running across the larva's field of view (bar width: 9°, speed: 20° s⁻¹, direction: randomly chosen from 12 angular directions 30° apart for each individual stimulus presentation interval, see polar plot inset). Scale bar = 100 μm (np: tectal neuropil, SPV: *stratum periventriculare*). **b** Average fluorescence modulations of all voxels in six anatomically identified zones during visual stimulation with moving bars (*n* = 8 larvae, color code: 1. dark blue = superficial contralateral neuropil, 2. blue = deep contralateral neuropil, 3. light blue = contralateral SPV, 4. yellow = ipsilateral SPV, 5. orange = deep ipsilateral neuropil, 6. red = superficial ipsilateral neuropil, dark gray shading indicates the 95% confidence intervals, light gray vertical bars indicate the stimulus interval). In addition to the expected stimulus-synchronized activity in the tectal hemisphere contralateral to the stimulated eye, visually evoked activity was also observed in the neuropil of the ipsilateral tectum. Moving bar directions for each stimulus presentation interval are indicated by black arrows above the traces with respect to a larva's orientation as shown in the polar plot inset in **a**. Source data are provided as Source Data File. **c** Highly stimulus-correlated voxels ($R^2 > 0.4$) were found in the tectal hemisphere contralateral to the stimulated eye, consistent with direct contralateral retinal input. However, stimulus-correlated voxels were also observed in the ipsilateral tectal hemisphere, which in this experiment does not receive any direct retinal input due to eye enucleation. Scale bar = 50 μm. Source data are provided as Source Data File. **d** The average density projection of voxels (calculated as the mean over the short axis of the curved rectangle in the left panel) that are highly correlated with a moving bar stimulus ($R^2 > 0.4$, as in **c** shows an enrichment in the tectal hemisphere contralateral to the stimulated eye and in the deep ipsilateral neuropil. Anatomical regions were color-coded as in **a** and **b** (*n* = 8 larvae, gray shading indicates the 95% confidence interval, np: tectal neuropil, SPV: *stratum periventriculare*, ML: midline).

that they must receive direct or indirect visual input. ITNs thus match the anatomical and functional profile of the commissural neuronal population whose existence was predicted from ipsilateral activity in the deep neuropil of the OT.

**ITNs are GABAergic interneurons and form synapses in the OT.** Next, we analyzed the neurotransmitter phenotype and synaptic connections of ITNs. We performed fluorescent whole-mount in situ hybridization in combination with anti-GFP immunostaining in 4 dpf *ITNGal4, UAS:GFP* zebrafish larvae. Expression of *vglut2a/2b* did not show any overlap with anti-GFP immunostaining suggesting that ITNs are not glutamatergic (Supplementary Fig. 3a). We furthermore found that ITN nuclei were located in the mesencephalic tegmentum, anterior to the midbrain–hindbrain boundary, and showed no overlap with *chata*-positive cells of the nucleus isthmus, in rhombomere 1 (Supplementary Fig. 3b)[2]. However, expression of *gad65/67*

colocalized with anti-GFP labeling in the ITN nuclei, indicating that ITNs are GABAergic inhibitory interneurons (Fig. 3a).

To find out where ITNs form synapses, we genetically targeted the presynaptic marker synaptophysin, conjugated to GFP (Syp-GFP)[18], to ITNs. We discovered that ITNs establish putative presynaptic contacts on their arbors in both tectal hemispheres. Presynaptic puncta were localized to the deep neuropil layers (Fig. 3b, upper right panel and Supplementary Fig. 2a, b), extending along the rostral-caudal axis of the OT (Fig. 3b, lower right panels). Furthermore, we transiently expressed the post-synaptic marker psd95-GFP[19] in ITNs and found putative postsynaptic specializations in the deep layers of the neuropil, ipsilateral to ITN cell bodies (Fig. 3c).

In summary, ITNs are a class of inhibitory interneurons that interconnect the two tectal hemispheres.

**ITNs respond to prey-like moving target stimuli.** In order to explore if ITNs are involved in visual processing during hunting

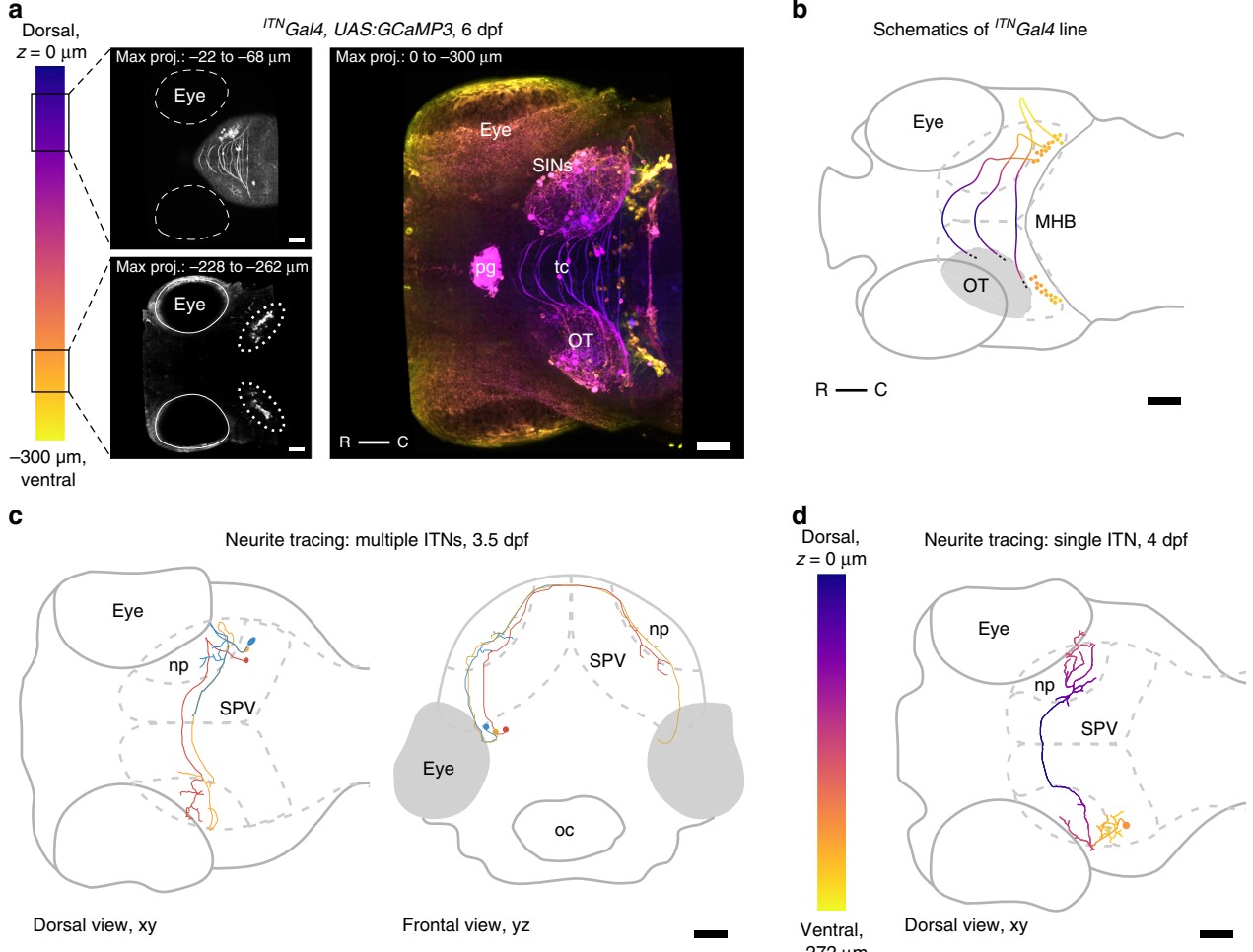

**Fig. 2** Intertectal neurons connect the two hemispheres of the optic tectum. **a** Maximum intensity projections of an $^{ITN}Gal4$ transgenic larva viewed dorsally at 6 dpf in which neurons that connect the two hemispheres of the optic tectum (intertectal neurons, ITNs) are labeled (most dorsal plane through the larva is indicated by $z = 0 \, \mu m$). The cell bodies of ITNs are situated in two bilateral symmetric nuclei below the respective tectal lobes (ITN nuclei highlighted by dotted ellipses in lower left panel) and send their axons dorsally through the tectum crossing the tectal commissure in ladder-like trajectories (upper left panel). In addition to labeling ITNs in the mesencephalic tegmentum, $Gal4$ is also expressed in the pineal gland, SINs, scattered periventricular neurons (PVNs), and the spinal cord in this transgenic line. All scale bars = 50 μm. (pg: pineal gland, tc: tectal commissure, OT: optic tectum, R: rostral, C: caudal, SINs: superficial interneurons). **b** Schematic of the transgenic $^{ITN}Gal4$ line depicted in **a**. ITN cell bodies and axon tracts are color-coded according to the position in the dorsoventral $z$-plane. To increase readability only the right ITN's connectivity with respect to the larva is shown. Scale bar = 50 μm. (MHB: midbrain-hindbrain boundary, OT: optic tectum, R: rostral, C: caudal). **c** Neurite tracing of multiple ITNs in a larva at 3.5 dpf. At 3.5 dpf ITN neurites start to cross the midline (e.g., the blue ITN). In addition, ITNs begin to form arbors at the boundaries between the deep layers of the neuropil and the PVN layer. Scale bar = 50 μm. (oc: mouth/oral cavity, np: tectal neuropil, SPV: *stratum periventriculare*). Color-coded to simplify distinction between single ITNs. **d** Neurite tracing of a representative ITN at 4 dpf, color-coded according to the position in the dorsoventral $z$-plane (most dorsal plane of the larva is indicated by $z = 0 \, \mu m$). ITN axons cross the midline superficially and form arborization patterns of increasing complexity in the ipsi- and contralateral neuropil structures of the optic tectum. Scale bar = 50 μm. (np: tectal neuropil, SPV: *stratum periventriculare*).

behavior, we first tested if ITNs respond to prey-like stimuli in a virtual hunting assay for tethered zebrafish larvae combined with 2-photon calcium imaging[13]. We presented 5–6 dpf $^{ITN}Gal4$, $UAS:GCaMP3$ larvae with small moving spots (speed: 30° s⁻¹, size: 5°) running horizontally across the frontal visual field (Fig. 4a) while simultaneously imaging ITN activity bilaterally. ITNs were strongly responsive to prey-like stimuli (Fig. 4b). Plotting the angular position of the moving spot at the onset of the calcium transient revealed that ITNs responded when the stimulus was in the contralateral visual hemifield. This is consistent with ITNs receiving visual input deriving from the contralateral eye via RGC terminals in the ipsilateral OT, or ipsilateral tectal interneurons (Fig. 4c). Simple light flashes did not evoke ITN responses and we did not observe activity modulations associated with convergent saccades (i.e., initiation of

hunting). This suggests that ITNs do not display premotor activity but rather visual sensory responses to prey-like moving target stimuli.

**ITNs are required for the binocular localization of prey**. Based on their anatomy and physiology, we hypothesized that ITNs might be required for the binocular localization of prey during hunting behavior. We examined this hypothesis using two loss of function approaches:

We generated $^{ITN}Gal4$, $UAS: BoTxBLC$-$GFP$ larvae to genetically silence vesicle release in $Gal4$-expressing neurons using zebrafish-optimized Botulinum toxin light chain[20], and assessed prey capture performance[9]. $^{ITN}Gal4$, $UAS: BoTxBLC$-$GFP$ larvae showed a large reduction in prey consumption compared with

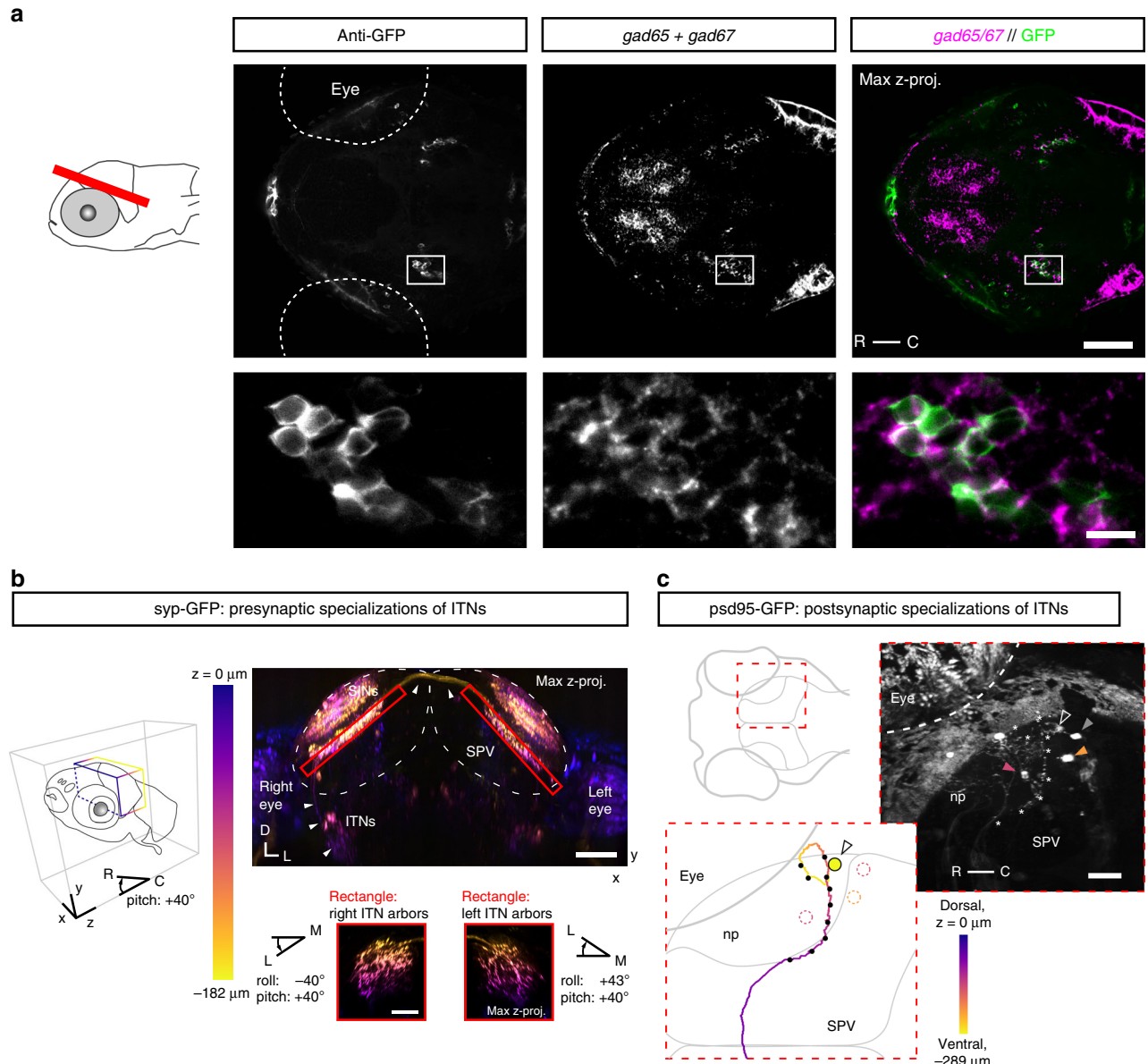

**Fig. 3 Intertectal neurons are GABAergic interneurons and form pre- and post-synaptic specializations in the OT. a** Expression of *gad65/gad67* and anti-GFP immunoreactivity in the brain of a 4 dpf *ITNGal4, UAS:GCaMP3* larvae (red bar in larva to the left indicates microscopic plane, scale bar = 50 μm). Two bilateral symmetric ITN nuclei are labeled by anti-GFP (left nucleus enlarged and shown in the lower images, scale bar = 10 μm). **b** Maximum intensity projection through the OT of a 5 dpf *ITNGal4, UAS:GCaMP3, UAS:Syp-GFP)* larva (view from the front and below of the larva). Syp-GFP signal is color-coded according to depth along *z*. Pre-synaptic specializations of ITNs (Syp-GFP puncta, red rectangles) are in the deep neuropil ipsi- and contralaterally to the ITN cell bodies. Small panels: maximum intensity projections of the pre-synaptic specializations of ITNs viewed dorso-laterally (larva first rotated along the roll axis by −40° for right ITN arbors or along the roll axis by 43° for left ITN arbors, then rotated along the larval pitch axis by +40°, color code according to depth in *z*). White arrows indicate the ITN axon tract. Syp-GFP-labeled puncta in upper layers of the tectal neuropils belong to superficial interneurons (SINs) in the SO and SFGS[54]. All scale bars = 50 μm. **c** Postsynaptic specializations labeled in a 3 dpf *ITNGal4 UAS:GCaMP3* larva, transiently expressing *UAS:psd-95-GFP*. Right panel shows a maximum intensity projection viewed dorsally through the right OT (skin autofluorescence partially removed). A single ITN was labeled in this larva (cell body indicated by open white arrow) and locations of puncta with strong psd-95-GFP expression are indicated by white asterisks along the ITN neurite. The ITN was traced and its trajectory color-coded according to the position in the dorsoventral *z*-plane (lower-left inset). Black dots along the neurite indicate the positions of psd-95-GFP puncta. Post-synaptic specializations of ITNs were found predominantly in the deep layers of the tectal neuropil ipsilateral to the ITN cell body. Periventricular neuron cell bodies (PVNs) were sometimes labeled in *ITNGal4* larvae (dotted circles or filled arrows color-coded according to their *z*-position in right panel). Scale bar = 30 μm. (np: tectal neuropil, R: rostral, C: caudal, D: dorsal, L: left, SO: *stratum opticum*, SFGS: *stratum fibrosum et griseum superficiale*, SPV: *stratum periventriculare*).

sibling controls (Supplementary Fig. 4a). However, further analysis revealed that *BoTxBLC-GFP*-expressing larvae did not inflate their swim bladder and thus spent significantly less time swimming and covered less distance during prey capture trials (Supplementary Fig. 4b). Therefore, reduced prey consumption is

likely secondary to locomotor deficits, perhaps due to *BoTxBLC-GFP* expression in the spinal cord.

Next we performed 2-photon laser ablation of ITN cell bodies to specifically examine the behavioral requirement of these neurons. We unilaterally ablated ITNs in 4 dpf *ITNGal4, UAS:*

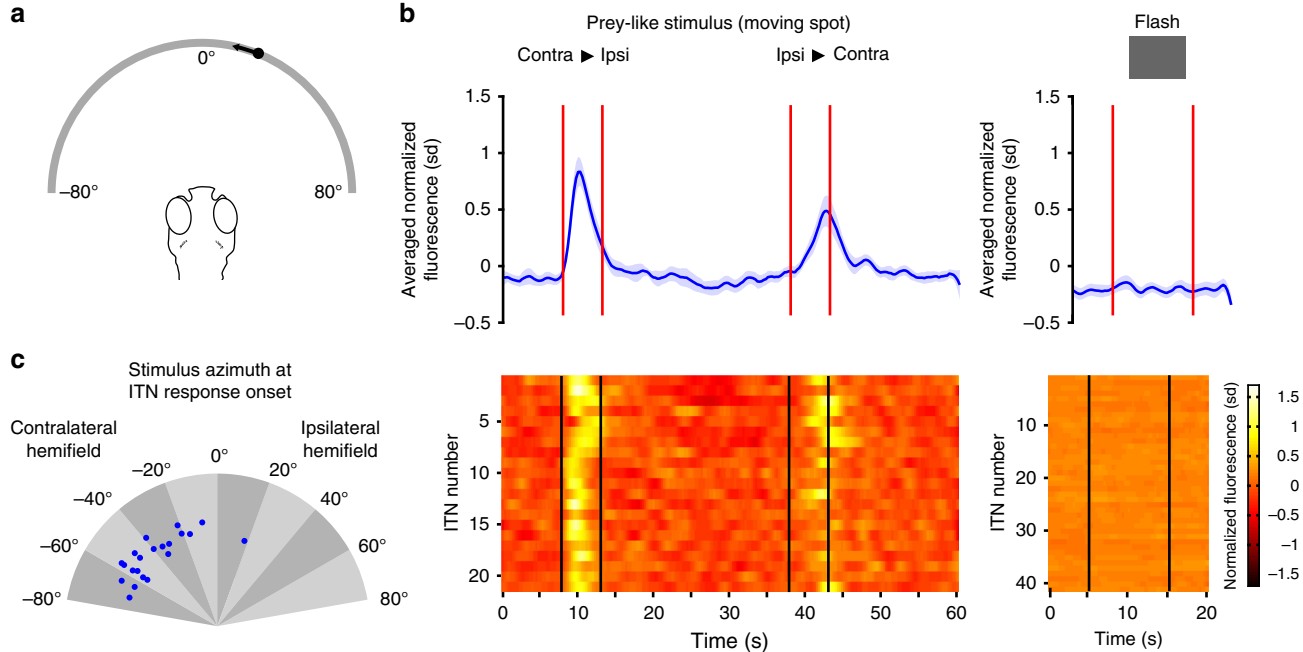

**Fig. 4** ITNs display visual sensory responses to prey-like moving target stimuli. **a** Schema of the virtual hunting assay. 5–6 dpf $^{ITN}Gal4$, $UAS:GCaMP3$ larvae were tethered in agarose with their eyes free to move. Small moving spots (size: 5°, speed: 30° s$^{-1}$) moving horizontally from right to left or vice versa were projected on a curved screen covering ~160° of visual space. At the same time, a 2-photon microscope was used to record fluorescent calcium signals in ITNs in response to the visual stimuli as well as eye kinematics using an infrared imaging camera. **b** ITN cell bodies showed strong Ca-transient modulations in response to moving spots (stimulus intervals indicated by the red/black vertical lines, $n = 21$ ITNs from five larvae) whereas ITNs did not respond to flashes ($n = 41$ ITNs from three larvae). Traces show mean normalized fluorescence intensities with 95% confidence intervals. Source data are provided as Source Data File. **c** Azimuth of moving spots at the onset of the response of each ITN to visual prey-like stimuli. ITNs collectively respond to moving spots spanning almost the whole contralateral visual hemifield ($n = 21$ ITNs from five larvae). Source data are provided as Source Data File.

$GCaMP3$ larvae (12–26 ablated ITNs, Fig. 5a) and, after a recovery period, assessed hunting performance of individual animals. Control larvae, of identical genetic background, underwent the same mounting and imaging procedure but were not ablated. As a further control, sham-ablations were performed by randomly ablating PVNs directly above the ITN nucleus. ITN-ablated larvae were imaged the next day, which confirmed that ITNs were successfully removed (Fig. 5a). At 6 dpf, control, sham-ablated and ITN-ablated larvae were tested in a prey capture performance assay that quantified consumption of paramecia over the course of 4 h.

ITN-ablated larvae consumed fewer paramecia than unablated controls (Fig. 5b) and the strength of this hunting deficit correlated with the number of ablated ITNs (Supplementary Fig. 4c). Sham-ablated larvae did not show a significant difference in paramecia consumption compared with control larvae and were thus pooled with the control group.

ITN-ablated larvae did not show differences in time spent swimming (Fig. 5c), swim bout duration, maximum bout speed, interbout interval or swim bout turn angles (Supplementary Fig. 4d–g), indicating that general motor defects do not account for the impairment of predatory performance. Moreover, prey recognition and hunting-specific motor outputs remained unchanged following ITN ablation: total hunting duration (as estimated from periods of ocular convergence) was unaffected (Fig. 5c) as was average hunting sequence duration and eye vergence angles during hunting (Supplementary Fig. 4h, i). Thus, despite capturing fewer prey than control larvae, ITN-ablated larvae do not display motor defects and can recognize prey and initiate hunting.

We further hypothesized that ITN ablation might result in reduced prey consumption if ITNs are required for larvae to estimate prey location and accordingly initiate capture swims. To uncover at which point during the hunting sequence ITN-ablated larvae deviated from normal hunting behavior, we analyzed a total of 2149 individual hunting sequences from 29 control and 12 ITN-ablated larvae. We subdivided individual hunting routines into a sequence of events enabling us to quantify probabilities of specific outcomes at each stage, including target switching, target fixation, successful capture swims etc. (Fig. 5d). Through this analysis, we detected a specific deficit at the penultimate fixation stage of a hunting sequence, immediately prior to striking at prey (Supplementary Movies 1 and 2). Specifically, when the prey was localized in the binocular strike zone (prey at 0.5 mm distance, ±10° azimuth), ITN-ablated larvae showed a significantly reduced probability of initiating a capture maneuver (Fig. 5g). By contrast, other aspects of hunting behavior of ITN-ablated larvae, including the probability and distance of target fixation, did not appear to differ versus controls (Fig. 5e, f and Supplementary Fig. 5a–f).

In summary, ablating commissural ITNs led to a specific behavioral deficit during binocular localization of prey[3], namely a failure to initiate capture strikes. This is consistent with reduced hunting performance in ITN-ablated larvae and supports the idea that ITNs mediate an interhemispheric neural computation that initiates capture swims at the final moment of hunting when prey is located in the binocular strike zone in front of the predator.

## Discussion

Ipsilateral retinal projections (IRPs) exist in members of almost all vertebrate classes. Their presence in basal cyclostomes suggests that IRPs are an ancestral trait[21] and seem to have been lost at multiple times during evolution such that they are absent or

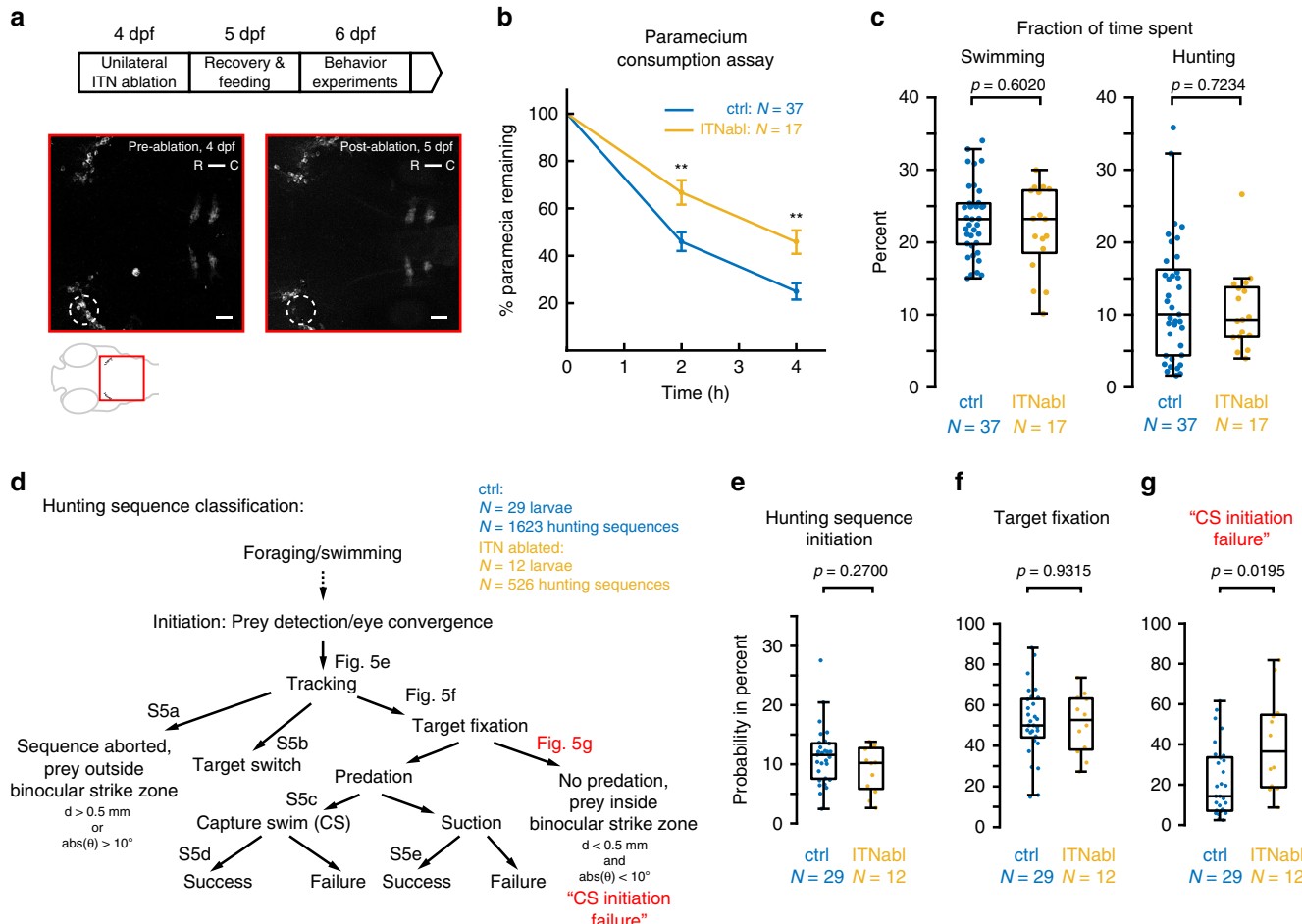

**Fig. 5** ITNs are required for the initiation of capture swims when prey is positioned in the binocular strike zone. **a** Experimental design for ablations and subsequent behavioral tracking. 12–26 ITNs were unilaterally laser-ablated in 4 dpf *ITN*Gal4, UAS:GCaMP3 larvae, left to recover and fed with paramecia on 5 dpf. Behavior experiments were performed at 6 dpf. ITN-ablated *ITN*Gal4, UAS:GCaMP3 larvae were imaged at 4 dpf (lower panels, pre-ablation) and again at 5 dpf (post-ablation). Dotted ellipse indicates ablated cells. Scale bars = 25 µm. **b** Paramecium consumption for control ($n = 34$ non-ablated larvae and $n = 3$ sham-ablated larvae) and ITN-ablated larvae ($n = 17$ larvae, 12–26 ITNs ablated). ITN-ablated larvae consume paramecia at a reduced rate compared with control larvae (percentage of remaining paramecia relative to the number of paramecia at $t = 0$ h, whiskers denote 95% confidence intervals, Mann–Whitney U-test, $t = 2$ h: $p = 0.0036$, $t = 4$ h: $p = 0.0013$). **c** Fraction of time spent swimming and time spent hunting for control ($n = 37$) and ITN-ablated ($n = 17$) larvae (Mann–Whitney U-test, swimming: median (ctrl) = 23.2%, median (ITNabl) = 23.2%, $p = 0.6020$//hunting: median (ctrl) = 10%, median (ITNabl) = 9.3%, $p = 0.7234$). **d** Schematic illustrating classification of individual hunting sequences. Figure references refer to the respective panels in Fig. 5 and Supplementary Fig. 5. In total, 1623 hunting sequences from 29 control larvae and 526 hunting sequences from 12 ITN-ablated larvae were analyzed. **e** ITN-ablated larvae initiate hunting with comparable probability to control larvae (control: median hunting sequence initiation probability = 11.6 %, ITN-ablated: median hunting sequence initiation probability = 10.2%, Mann–Whitney U-test, $p = 0.2700$). **f** ITN-ablated larvae fixate targets with a comparable probability to control larvae (control: median target fixation probability = 50.0%, ITN-ablated: median target fixation probability = 52.7%, Mann–Whitney U-test, $p = 0.9315$). **g** ITN-ablated larvae initiate capture swims at lower probability when prey is positioned in the binocular strike zone (prey at <0.5 mm distance, ±10° azimuth) compared with controls (Mann–Whitney U-test, $p = 0.0195$, control: median failure rate = 14.3 %, ITN-ablated: median failure rate = 36.5 %). Central marks of boxplots indicate the respective median. Bottom and top edges correspond to 25th and 75th percentiles. Source data for Fig. 5b–g are provided as Source Data File.

reduced in certain species (for a comprehensive review see ref.[22]). The existence of IRPs is sometimes attributed to a high degree of overlap of the field of views from both eyes and/or a predatory life style relying on binocular stereoscopic vision. However, this cannot be considered a general rule since there are species that have extensive IRPs but no overlapping field of views (e.g., the hagfish *Eptatretus*)[21] as well as predators such as the chameleon, which lack IRPs entirely[23].

Zebrafish larvae are predatory, also lack IRPs to the OT[11] but show behavioral responses consistent with processing of binocular cues during prey hunting[3,8,9]. In this study, we have presented evidence for an intertectal neural circuit that could enable a vertebrate without IRPs to process binocular visual cues.

To our surprise, 2-photon calcium-imaging in zebrafish larvae revealed visual motion-evoked neuronal activity in the deep layers of the tectal neuropil ipsilateral to the stimulated eye, despite the lack of ipsilateral retinotectal projections. We identified a Gal4-expressing transgenic line which labels a previously unknown class of commissural neurons (ITNs) that connect both tectal hemispheres. ITNs respond to visual motion stimuli in the contralateral visual field and extend arbors in the deep neuropil laminae of the OT where ipsilateral visual activity is observed. It is therefore likely that ITNs transfer visual information across the midline, which in turn would enable tectal integration of visual information from the two eyes. Unilateral ablation of ITNs impaired prey hunting performance and produced a specific deficit in the initiation of capture

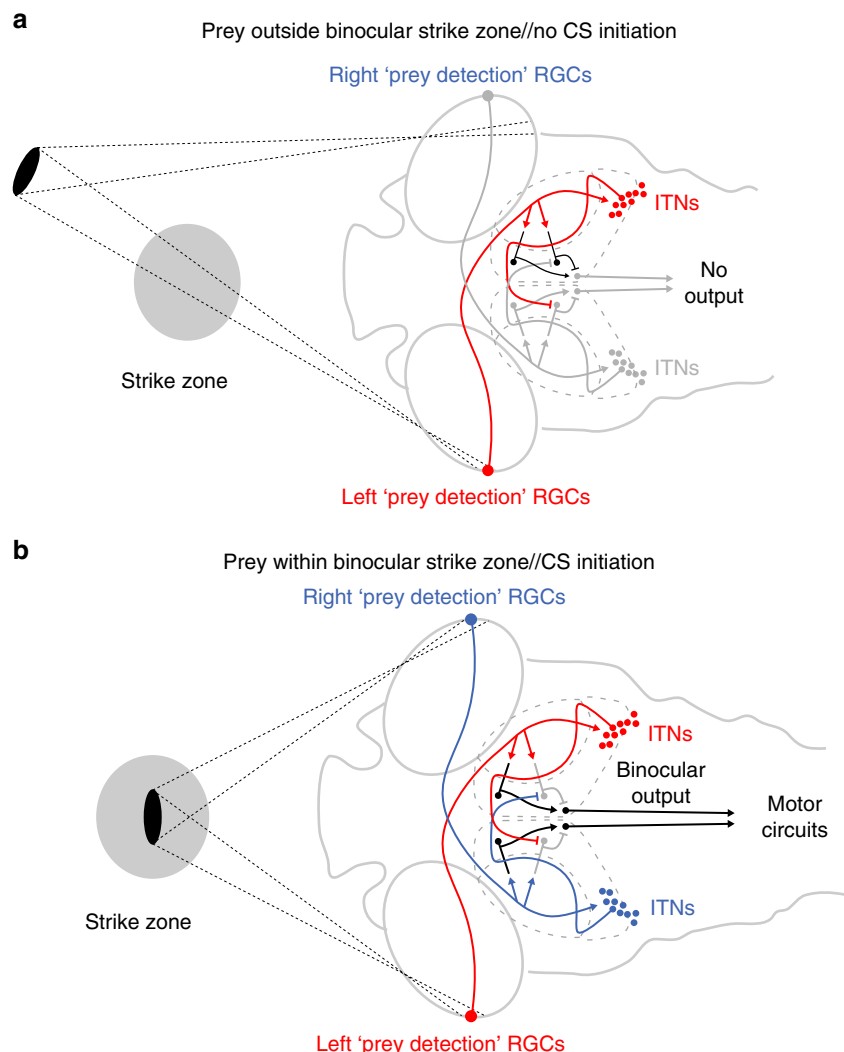

**Fig. 6** Model circuit for the integration of binocular visual input to localize prey and initiate capture swims. **a** Prey located outside the strike zone will at most activate the trigger zone of just one eye. RGCs transmit visual information to the contralateral OT and contralateral ITNs. ITNs in turn cross the midline to convey visual information to the opposite OT. In this case, local inhibitory interneurons in the opppsite OT prevent a capture swim from being triggered. **b** When prey is positioned inside the binocular strike zone, both trigger zones are activated. Thus, each tectum receives direct contralateral retinal input, as well as indirect, ITN-mediated input carrying information from the ipsilateral eye. The coincidence of excitatory drive and disinhibition, respectively, allows a premotor command to be generated to initiate a capture swim.

swims. Although ITN-ablated larvae continued to fixate prey at equivalent distances compared to controls, they failed to initiate capture maneuvers when prey was localized to the binocular strike zone. The presence of intertectal connections in zebrafish larvae was previously suggested based on anatomical data from immunohistochemistry[24,25] or single cell labeling approaches[26,27]. However, our study represents the first anatomical, physiological, and behavioral assessment of a genetically accessible class of commissural intertectal neurons.

At the penultimate fixation stage of hunting epochs, eye vergence angles are almost invariant[3]. Thus, lines of sight to the binocular strike zone will be represented at consistent locations in temporal retinae of the two eyes in accordance with anatomical evidence showing increased photoreceptor density in the temporal retina of larval zebrafish[28,29]. Coincident activation of these retinal trigger zones could be the neural signal indicating the correct localization of prey that in turn is used to initiate a capture swim[3]. This simple triangulation mechanism is reminiscent of that proposed for the striking behavior of preying mantids[30,31].

In particluar, we suggest that ITNs participate in a circuit that detects the presence of prey in the binocular strike zone by signaling coincident, bilateral activation of specialized trigger zones in the left and right temporal retinae (Fig. 6). In our model, retinal activity is relayed to contralateral tectal neurons as well as contralateral ITNs, consistent with ITN activation by prey-like stimuli in the contralateral visual field (Fig. 4). ITNs in turn project across the midline (Fig. 2) to transfer visual information to the opposite hemisphere (Fig. 1). As ITNs are inhibitory (Fig. 3), we propose a disinhibition mechanism whereby they silence inhibitory interneurons in the contralateral tectum. This enables a circuit motif in which the tectum can detect simultaneous activation of both trigger zones: activation of the contralateral retina is communicated by direct retinal input, and activation of the ipsilateral temporal retina is signaled by ITN-mediated disinhibition of intrinsic interneurons. The combination of excitatory drive and removal of inhibitory block together allow the generation of a pre-motor command that triggers execution of a capture swim.

Binocular neural responses have been previously reported in zebrafish larvae in the context of behaviors driven by whole field motion: the optokinetic reflex (OKR) and the optomotor response (OMR)[32–34]. Here convergence of monocular visual information is thought to be mediated via the posterior commissure[33] and because these behaviors do not depend on retinotectal connections[15] it is likely that binocular integration of whole field motion cues is independent of ITNs.

Intertectal connections have been studied in multiple species[35,36]. In anurans, neurons of the *nucleus isthmi* indirectly connect the tectal hemispheres and mediate binocular visual responses in the anterior part of the OT[37,38]. However, bilateral ablation of these nuclei does not impair binocular depth perception during hunting[37,39]. Direct commissural connections between the superior colliculi, the mammalian homolog of the OT, were also reported in mammals but their function remains rather obscure: Severing the collicular commissure restored visual fixation capabilities of hemianopic cats to the blind hemifield that was originally caused by unilateral ablations of the visual neocortex (Sprague effect)[40]. While the interpretation of this experiment is not straight forward, it suggests a mostly underappreciated role of intercollicular neurons in visually guided behavior, even in mammals.

Finally, it has been convincingly shown that binocular stereopsis is present in very different evolutionary lineages[41–43]. However, common and distinct principles of binocular stereopsis have not been addressed yet on the neural circuit level. Our study contributes to defining these principles by providing evidence for a neural circuit that establishes a simple form of binocular stereopsis in the absence of direct superposition of retinal input from both eyes.

## Methods

**Contact for reagent and resource sharing**. Further information and requests for resources and reagents should be directed to and will be fulfilled by Filippo Del Bene (filippo.del-bene@inserm.fr).

**Zebrafish embryo maintenance**. Zebrafish (*Danio rerio*) were maintained at 28 °C on a 14 h light/10 h dark cycle. Fish were housed in the animal facility of our laboratories at UCL or Institut Curie, which were built according to the respective local animal welfare standards. All animal procedures were performed in accordance with French, British, and European Union animal welfare guidelines. Animal handling and experimental procedures were approved by the Committee on ethics of animal experimentation—Institut Curie and the UK Home Office under the Animal (Scientific Procedures) Act 1986. No statistical methods were used to predetermine sample size. Where indicated the experiments were randomized and the investigators blinded to allocation during experiments and outcome assessment.

**ITN neurite and synapse labeling**. Transient single cell labeling to analyze ITN neurite morphology was achieved by injecting 1 nl of a *UAS:tagRFP, UAS:GFP* plasmid DNA (25 ng μl⁻¹) at 1 cell stage into *ITNGal4, UAS:GCaMP3* embryos. To transiently label post-synaptic densities of ITNs, 1 nl of a *UAS:psd95-GFP* plasmid DNA (25 ng μl⁻¹) were injected at 1 cell stage into *ITNGal4, UAS:GCaMP3* embryos. Presynaptic sites were labeled by crossing *ITNGal4, UAS:GCaMP3* with *Tg (Brn3c:Gal4, UAS:Syp-GFP)* larvae and the offspring screened against *Brn3c* and for *Gal4ic3034Tg*—and *Syp-GFP* expression. Larvae were imaged at 3–5 dpf using a Zeiss LSM 780 confocal microscope (Zeiss, Oberkochen, Germany), equipped with a water immersion objective (Zeiss, Plan-Apochromat 40 ×/1.00). Image stacks were analyzed using MATLAB (The MathWorks Inc., Natick, MA, USA) and FIJI[44]. Skin autofluorescence was removed manually in FIJI to enable an unobstructed view onto ITN synaptic structures and neurites.

**Neurite tracing**. ITN neurites were traced using the simple neurite tracer plugin[45] provided in FIJI. Individual ITNs labeled in *ITNGal4, UAS:GCaMP3* larvae injected with *UAS:tagRFP, UAS:GFP* were followed through the confocal stack based on their color and/or marker intensity. The vector files containing the traced neurites were then exported to MATLAB for further processing using the MatlabIO toolbox included in visualization and analysis software Vaa3D[46].

**Whole-mount in situ hybridization and immunohistochemistry**. *Vglut2a/vglut2b*, *gad67/gad65* probe mixtures were generated from cDNA using the following primers:

vglut2a_fwd: GATTCTCCTCACGTCCACACTGAA
vglut2a_rev: AACACATACTGCCACTCTTCTCGG
vglut2b_fwd: CGTCGACATGGTCAATAACAGCAC
vglut2b_rev: ATAGCACCTACAATCAGAGGGCAG
gad67_fwd: CATCATCCTCACCAGCTGCTGGAG
gad67_rev: AACATTGTAAAGGCACACCCATCATC
gad65_fwd: TCACCTATGAGGTGGCTCCAGTCTTC
gad65_rev: GTCATAATGCTTGTCCTGCTGGAAC

The mRNA anti-sense riboprobes for *chata* were kindly provided by Marnie Halpern[47]. For in situ hybridization, larvae were stored in 100% methanol and rehydrated stepwise into 0.1% Tween in PBS. Tissue permeabilization was achieved by proteinase K treatment adapted to larval age at room temperature, followed by post-fixation in 4% PFA in PBS. After prehybridization, hybridization with digoxigenin-UTP-labeled and fluorescein-UTP-labeled riboprobes (Roche Applied Science) was performed overnight at 65/68 °C and larvae were kept in the dark for all subsequent steps. Larvae were washed with TNT (0.1 M Tris, pH 7.5, 0.15 M NaCl, 0.1% Tween 20), incubated in 1% H₂O₂ in TNT for 20 min, washed several times and blocked in TNB [2% DIG Block (Roche) in TNT] for 1 h. Incubation with anti-digoxigenin-POD (peroxidase) Fab fragments (Roche, 1:50 in TNB) was performed for 24 h at 4 °C. Signals were detected using the Cy3-TSA kit (PerkinElmer) for 60 min (*vglut2a/2b* mixture) or 30 min (*gad65/67* mixture), respectively. Larvae were incubated in DAPI in TNT overnight at 4 °C and washed with TNT prior to imaging. For subsequent immuno-staining, chicken anti-GFP (1:500, Genetex, GTX13970) and mouse anti-ERK (1:500, p44/42 MAPK (Erk1/2), 4696) were used as primary antibodies followed by goat anti-chicken Alexa Fluor 488-conjugated (1:400, Invitrogen, A11039) and anti-mouse Alexa Fluor 594-conjugated (1:500, Invitrogen, A11005) secondary antibodies.

**3D Brain registration**. Registration of image volumes of 18 *ITNGal4, UAS:GCaMP3* at 6 dpf, co-immunolabeled with anti-tErk and anti-GFP, to a tErk reference brain[48] was performed using the CMTK toolbox version 3.2.2[49].

**Eye ablations**. 3 to 4 dpf *Tg(elavl3:GCaMP5G)* larvae were mounted in 2% low-melting-point agarose and one eye was removed using fine forceps. The larvae were then left to recover in Ringer solution supplemented with Calcium for 1 h. Subsequently, the larvae were transferred into fish medium until 2-photon Calcium imaging was performed on 5 or 6 dpf.

**Lipophilic dye tracing**. For tracing of retinotectal projections, eye-ablated larvae where fixed in 4% PFA/PBS for 2 h following injection of the lipophilic dye DiO into the remaining eye.

**Calcium imaging and visual stimulation**. Larvae were mounted in 2.5% low melting-point agarose such that one eye was facing an OLED screen (800 pixel × 600 pixel, eMagin, Bellevue, WA, USA) subtending around 70° × 55° of a larva's visual field. The screen was covered with a red long-pass filter (Kodak Wratten No. 25) to enable simultaneous imaging and visual stimulation. Larvae were then imaged in vivo using a two-photon microscope (LaVision Biotec, Bielefeld, Germany), equipped with a mode-locked Chameleon Ultra II Ti–Sapphire laser tuned to 920 nm (Coherent Inc., Santa Clara, CA, USA) and a Zeiss Plan-APOCHROMAT 20× water immersion objective (NA 1, Zeiss, Oberkochen, Germany). Emitted fluorescence was detected using a bandpass filter (ET525/50 M, Chroma Technology GmbH, Olching Germany) in front of a GaAsP photomultiplier tube (H7422-40, Hamamatsu Photonics K.K., Hamamatsu City, Japan). The average laser power at the sample during scanning was 10–20 mW and frames were scanned at 4 Hz resulting in 1.33 μs dwell time per pixel (pixel size: 0.85 μm²). Scanning and image acquisition were controlled with LaVision Biotec's proprietary ImSpector software.

Visual stimulation was handled by a computer separate from the image acquisition setup using Psychophysics Toolbox[50]. Stimuli were either moving bars running across the eye's field of view (bar width: 9°, speed: 20° s⁻¹, direction: pseudo-randomly chosen for each stimulus epoch from 12 angular directions 30° apart, stimulus repetitions: three times per direction, stimulus presentation interval: 10 s) or light flashes instantaneously covering the entire field of view (stimulus repetitions: five times, stimulus presentation interval: 10 s). The image acquisition was synchronized with the visual stimulation by recording the imaging frame number in parallel to stimulus presentation timing via a U3 LabJack DAQ (LabJack, Lakewood, CO, U.S.A.).

**Virtual prey-hunting assay**. 5 to 6 dpf *ITNGal4, UAS:GCaMP3* larvae were individually mounted in 2% low-melting-point agarose in a 35 mm petri dish and the eyes and tail subsequently freed from the agarose with a scalpel. Visual stimuli consisted of small moving spots (subtending 5° visual angle) back-projected onto a curved screen in front of the animal, appeared at 76° to the left or right of the midline and then moving 152° right or left across the frontal region of visual field (at an average angular velocity with respect to the fish of 30°/s). At each imaging

plane, 12–18 repetitions of each of the visual stimuli (L > R, R > L) were presented in pseudo-random order.

2-photon calcium imaging was simultaneously performed using a custom-built microscope equipped with a 20 × /1.00 NA Olympus objective and a Ti:Sapphire ultra-fast laser (Chameleon Ultra II, Coherent Inc) tuned to 920 nm. Average laser power at the sample was 5–10 mW. Images were acquired by frame scanning at 3.6 Hz with 1 μs dwell time per pixel. For each larva, 1–3 planes through the ITN nuclei of the larva were imaged. Image acquisition and visual stimulus presentation were controlled using software written in LabView and MATLAB.

**Image analysis for calcium imaging experiments.** A voxel-based analysis strategy was chosen for the Calcium-imaging experiments following eye ablations. Acquired timeseries were motion-corrected by calculating the shift between an average image and each timeseries image based on the inverse Fourier-transformed normalized cross-power spectra. Each image timeseries was further registered to the corresponding z-plane of a 5 dpf Tg(elavl3:GCaMP5G) reference larva using MATLAB's align_nonrigid function. Individual image timeseries were re-shuffled due to the randomized stimulus order during visual stimulation and averaged across repetitions. To identify voxels that were correlated with stimulus presentation we used a regression-based approach[16]. In short, a binary matrix was generated indicating when a stimulus was present independent of its directionality. For every voxel, the average temporal response profile was estimated using least-squares regression. Voxels with a p value > 0.0001 were discarded from analysis and the maximum value of the regression coefficient from eight larvae was kept for each voxel. Voxels that had a regression coefficient > 0.4 were considered strongly correlated with stimulus presentation and thus active. The intensity values for all voxels in one of six anatomically defined ROIs in the ipsilateral and contralateral tectal hemisphere were averaged over time for each larva. The deltaF/F0 average profiles were calculated by estimating the baseline using MATLAB's imerode function with a 10 s kernel (i.e., the stimulus presentation interval). DeltaF/F0 was obtained by calculating $F_{<ROI>}/F_{<ROI\ baseline>} - 1$. DeltaF/F0 average profiles were then averaged across larvae.

The response of ITNs to moving bars and light flashes covering the contralateral field of view was analyzed using a ROI based approach. First, acquired timeseries were registered by calculating the shift between an average image and each timeseries image based on the inverse Fourier-transformed normalized cross-power spectra. ROIs were manually selected from a standard deviation projection of the image timeseries and based on anatomical landmarks. ROI timeseries were re-shuffled to account for the randomized stimulus order during visual stimulation and averaged across repetitions. The deltaF/F0 average profiles were calculated by estimating the baseline using MATLAB's imerode function with a 10 s kernel and calculating $F_{<ROI>}/F_{<ROI\ baseline>} - 1$. Finally, the deltaF/F0 averages were normalized to the maximal value.

Analysis of data collected during virtual hunting assays was performed by correcting motion artefacts of timeseries using cross-correlation[13]. ROIs were selected manually based on an anatomical stack of the brain volume. ROI trials were discarded from analysis if the z-scored motion-error exceeded two standard deviations. The ROI intensity profiles were z-scored and then averaged over trials. Lateral ITN position (i.e., left or right) was recorded from an anatomical stack of the larva and stimulus direction (right to left and vice versa) was remapped to contralateral to ipsilateral and vice versa with respect to the ITN position in the larva. For calculating the onset of the ITN activity with respect to the stimulus position in the visual field, the maximum of the ROI timeseries within the stimulus interval (contra > ipsi and ipsi > contra) was identified. Subsequently, the response onset was defined as timepoint when the response curve was smaller than the maximum and falling below 0.5 standard deviations. Timepoints for contra > ipsi and ipsi > contra were averaged and the corresponding azimuth was calculated based upon the trajectory of the visual cue.

**Synaptic silencing.** ITNGal4, UAS:RFP larvae were crossed to UAS:BoTxBLC-GFP larvae[20], screened at 3 dpf and separated into GFP-positive (ITNGal4, UAS: BoTxBLC-GFP) and GFP-negative (control) siblings. On 5 dpf, all larvae were fed with rotifers to gain feeding experience.

**ITN laser ablations.** At 4 dpf, ITNGal4, UAS:GCaMP3 larvae, in which more than 12 ITNs were labeled in one nucleus, were randomly assigned to either the control, sham-ablated or ITN-ablated group. Larvae from the two ablation groups were then randomly selected for unilateral cell ablation (i.e., only cells on either the left or right side with respect to the larval rostral–caudal axis to be ablated per larva). Then all larvae independent of the assigned group were anesthetized using 0.02% tricaine (MS-222, Sigma) diluted in fish medium, mounted in 3% low-melting-point agarose and imaged using a custom-made 2-photon microscope[13]. Single target cells (either ITNs or PVNs) were identified by first taking a full frame scan. Then a mode-locked laser beam (800 nm) was scanned in a spiral pattern around a defined target cell position for ~100 ms. A cell was regarded as successfully ablated when after ablation a small point of saturated intensity was detected at the target position instead of the cell. Then the procedure was repeated and finally the number of successfully ablated cells recorded for each larva. After the procedure, each larva's brain was inspected under the microscope and only larvae with clearly

visible heartbeat and blood flow in both brain hemispheres were kept for further experiments. Control, sham-ablated and ITN-ablated larvae were then removed from the agarose and kept individually. On 5 dpf ITN-ablated larvae were re-imaged to ensure that neurons were successfully removed and were otherwise left to recover. All larvae were subsequently separated into 12-well plates and assigned a code for anonymization. Each larva was fed with Paramecia on 5 dpf to gain feeding experience.

**Prey consumption assays.** At 6 dpf, all larvae were tested in a prey consumption assay[9] and, since the larvae received food more than 12 h earlier, they were at this point considered starved[51,52]. At the beginning of the assay ($t = 0$ h) paramecia or rotifers were added to petri dishes containing either a single larva (for ITN-ablations) or 4 larvae (for the BoTxBL-GFP-expression experiments). Short movies of the dishes were then recorded at $t = 0$ h, $t = 1$ h, $t = 2$ h and $t = 4$ h after the start of the experiment using high-speed infrared sensitive cameras (MC1362 or MC4082, Mikrotron GmbH, Germany) positioned above the dish. Dishes were dark field-illuminated using a custom-made LED-ring (850 nm) placed around the dishes. To obtain temporal prey consumption curves, the number of living prey per dish was counted at each time point during the assay using FIJI's Cell Counter plugin. These numbers were then normalized to the initial prey number at $t = 0$ h.

**Larva tracking and data analysis.** For the behavioral analysis of UAS:BoTxBLC-GFP-expressing larvae, individual larvae were imaged with a Mikrotron MC1362 high-speed camera at 700 Hz. Each recording lasted between 5 to 10 min. The larva's position in each frame was extracted online using a custom-made tracking software[1].

For the analysis of ITN-ablated larvae, individual animals were filmed for around 15 min at 100 frames s⁻¹ and illuminated by a custom-made LED-based diffusive backlight (850 nm) placed below the dish. For each frame, the $x/y$ - position and orientation of the larva's body centroid and the angles of the larva's eyes were extracted in real-time based upon image moments computed by OpenCV routines within the open-source visual programming language Bonsai[53].

Data Analysis was performed off-line in MATLAB. Hunting sequences were detected by calculating the vergence angles from the difference between the left and right eye angles filtered by a 50 ms boxcar filter. Vergence distributions are mostly bimodal (i.e., representing eyes parallel and convergences) thus thresholds for start (on) and end (off) of hunting sequences were determined by fitting the vergence distributions for each larva with a Two-Term Gaussian Model (using MATLABs fit function). The on-threshold for hunting sequences for each larva was then determined by subtracting 1 standard deviation from the mean of the gaussian distribution with the higher mean (representing convergences). The off-threshold for hunting sequences was the intersection point of two gaussians minus 5°. The start of a hunting sequence was then defined by the point when the vergence angle exceeded the on-threshold and the end was indicated by the vergence angle falling below the off-threshold. From these hunting sequence start and end timepoints, the fraction of time spent hunting could be calculated by taking the number of video frames during hunting sequences normalized by the total number of frames during the experiment. Furthermore, hunting sequence duration and average vergence angle during hunting sequences were determined.

Swim bouts were detected by calculating a larva's instant velocity in mm s⁻¹ from the x- and y-centroid position difference filtered with a 30 ms (for determining the start of a swim bout) or a 100 ms (for determining the swim bout end) boxcar filter. Putative swim bouts were then detected using MATLABS findpeaks function. Start and end timepoints of swim bouts were determined based on the on-threshold of 2 mm s⁻¹ in the 30 ms-boxcar-filtered velocity curve and an off-threshold of 1.25 mm s⁻¹ in the 100 ms-boxcar-filtered velocity curve. From the swim bout start and end timepoints, the fraction of time spent swimming could be calculated by taking the number of video frames during swim bouts normalized by the total number of frames during the experiment. Furthermore, swim bout duration, maximum bout speed and interbout intervals were calculated for bouts and interbout intervals occurring during hunting sequences for each larva. To obtain turning angles for each bout during hunting sequences the difference between average orientation changes 30 ms before and 30 ms after a swim bout was calculated. For ITN-ablated larvae turning angles were re-mapped according to the side of the ablation to control for potential unspecific ablation defects induced through unilateral ablation.

For all analyzed hunting sequence and swim bout parameters, the relative frequency distribution and the cumulative relative frequency distribution per larva were calculated and then averaged for all larvae within the control and ITN-ablated group. To test for equality of the average distributions a Two-Sample Kolmogoroff–Smirnov test was performed.

**Single hunting sequence evaluation of ITN-ablated larvae and data analysis.** To evaluate the outcome of single hunting sequences, 15 min videos of hunting larvae were taken at 100 frames s⁻¹. Dishes were dark field-illuminated using a custom-made LED-ring (850 nm) placed around the dish to simplify the detection of paramecia. Hunting sequences were identified based on visible eye vergences and assigned to either of the categories shown in Fig. 5d based on their outcome: 1. sequence aborted when prey outside of binocular strike zone (prey at distance

smaller than 0.5 mm and smaller than ±10° azimuth between prey and larva), 2. targets were switched or 3. target was fixated i.e., the larva was positioning the target in a stereotypic binocular strike zone around 0.5 mm in front of the larva. If the target was fixated during the hunting sequence it was furthermore evaluated if a capture swim, a suction or sequence abortion when the target was inside the binocular strike zone occurred. Finally, if a capture swim or a suction was performed the success or failure was recorded. Only sequences with an unequivocally identifiable fixated target were included for analysis. In rare cases in which multiple paramecia were fixated by a larva during a single hunting sequence (43/2149 hunting sequences from 29 control and 12 ITN-ablated larvae), the entire hunting sequence was assigned to the target switch category. Data analysis was performed off-line in MATLAB. Category probabilities for a given larva were calculated by counting the total number of events (e.g., successful capture swims) normalized to the number of events of the superordinate node (e.g., capture swims).

**Quantification and statistical analysis**. All statistical analyses were performed in either MATLAB or FIJI. Statistical tests, p values and sample size are reported in the text or figure legends. All tests were two-tailed. No statistical methods were used to predetermine sample size. Where indicated the experiments were randomized and the investigators blinded to allocation during experiments and outcome assessment.

**Reporting summary**. Further information on research design is available in the Nature Research Reporting Summary linked to this article.

## Data availability

The datasets generated and analyzed during the current study are available from the corresponding authors on request. The source data underlying Figs. 1b, c, 4b, c, 5b–g and Supplementary Figs. 2d, 4a–i and 5a–f are provided as a Source Data file.

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

## Acknowledgements

We would like to thank Kristen Severi and Roshan Jain for comments on the manuscript and all members of the Del Bene and Bianco labs for discussions. The Tg(*UAS:BoTxBLC-GFP*) zebrafish line was a gift from Claire Wyart. We thank the Developmental Biology Curie imaging facility (PICT-IBiSA@BDD, Paris, France, UMR3215/U934), member of the France-BioImaging national research infrastructure, for their help and advice with everything microscopy related. The Del Bene laboratory Neural Circuits Development is part of the Laboratoire d'Excellence (LABEX) entitled DEEP (ANR -11-LABX-0044), and of the École des Neurosciences de Paris Ile-de-France network. This work has been in part funded by an ATIP/AVENIR program starting grant (F.D.B.), ERC-StG #311159-Zebratectum (F.D.B.), CNRS, INSERM and Institut Curie (F.D.B) core funding. I.H.B. received a UCL Excellence Fellowship and a Sir Henry Dale Fellowship (101195/Z/13/Z) from the Wellcome Trust and Royal Society. C.G. was supported by an EMBO Short-term Fellowship, an Institut Curie Postdoctoral Fellowship and an FRM Postdoctoral Fellowship. T.O.A. was supported by a Boehringer Ingelheim Fonds PhD Fellowship. P.H. was supported by a UCL IMPACT studentship. G.R. received funding from the European Union's Horizon 2020 Research and Innovation program under the Marie Skłodowska-Curie grant agreement no. 666003.

## Author contributions

C.G., I.H.B., and F.D.B. conceived and designed experiments. C.G. performed all experimental procedures unless otherwise indicated. T.O.A. and C.G. identified the *ITN*Gal4 line. P.H. performed the virtual hunting assay experiments. T.O.A. performed in situ hybridizations, morphological ITN characterization, and the synaptic labeling experiments. K.D. helped with immunostaining experiments and zebrafish line maintenance. G.R. performed the behavioral analysis of *ITN*Gal4 fish expressing BoTxBLC-GFP. C.G. analyzed the data. C.G., I.H.B. and F.D.B. wrote the manuscript with input from all coauthors.

## Competing interests

The authors declare no competing interests.
