## [Peer Review File · Nature Communications]

Reviewers' comments:

Reviewer #1 (Remarks to the Author):

The manuscript by Gebhardt et al. describes the characterization of a novel group of cells termed intertectal interneurons (ITNs), which mediate communication between the left and right tectal hemispheres of the larval zebrafish brain. Through calcium imaging and ablation experiments, the authors show that ITNs are motion responsive and important for the visualization and capture of prey. This is an inherently interesting study because it answers a long-standing question of how the fish visual system can functionally effectively to integrate binocular cues, when comprised of only contralateral and no ipsilateral retinal projections.

Despite my enthusiasm for this work, I find the organization of ideas somewhat awkward and the writing not always accessible. Use of poorly explained jargon and stacked modifiers do not help if this study is to be understood by a broad readership. These concerns can be easily rectified.

Specific issues:

1. As presented, the premise behind the study seems somewhat artificial – that the authors were examining ipsilateral activation of the tectum by visual stimuli, “predicted the existence of motion responding commissural neurons that connect the tectal hemispheres” and just happened to have a transgenic gene trap line in hand that labels the relevant, bilaterally projecting neurons. It would seem the other way around - that the authors recovered a zebrafish line with an unexpected axonal labeling pattern and explored what the function of the bilaterally projecting neurons might be. This seems far more plausible and, if accurate, should be presented in this manner.

2. The authors are strongly advised to use standardized nomenclature (<https://wiki.zfin.org/display/general/ZFIN+Zebrafish+Nomenclature+Conventions#ZFINZebrafishNomenclatureConventions-4.3>) for labeling their gene trap line. ITNGa4 is not correct according to accepted naming practice in the field. Moreover, seeing as how their new line labels a unique and poorly studied neuronal population, it is of interest to know the genetic locus where the transgenic insertion resides. This could be readily determined by inverse PCR.

3. Why were the ITN ablation experiments only performed unilaterally? Is there a difference in the behavioral responses if all of the ITNs are ablated?

4. The model depicted in Fig. 6 does not take into account that ITNs were found to be GABAergic and propose a plausible inhibitory role for these cells. Even in the Discussion, it is confusing that the authors argue that execution of prey capture behavior requires “an ITN-mediated disinhibition in both tectal hemispheres” (does this mean disinhibition of ITNs or ITNs disinhibiting other neurons?). One would think that activation of ITN GABAergic neurons would have an enhanced inhibitory effect. I don't follow the logic of p. 11, para. 2 to p. 12, para. 1, as presented. Moreover, since tools for optogenetic activation or synaptic inhibition of Gal4 expressing neurons are available, it would be beneficial to examine the behavioral consequences of modulating the activity of ITNs. Such experiments would greatly strengthen this study and help in framing the model.

5. A few sentences at the start of the Discussion reviewing the differences between vertebrate species in the extent and retention of ipsilateral retinal projections (and summarizing current models to account for this diversity) would be very helpful for a reader to gauge the broader significance of this study. Although there is some mention of the potential existence of similar neurons in other species, it would strengthen this study if the authors could make a better case for conservation of ITN-like neuronal function.

Minor suggested changes:

Abstract

p. 2, l. 39, modify to "a novel commissural neuronal population termed intertectal interneurons (ITNs) that connect the tectal hemispheres"

Introduction

p. 3, l. 53. Need a better transition sentence between paragraphs 1 and 2 to clearly state that fish have laterally positioned eyes. The issue is not clearly explained here- too cryptic for a general readership as written.

p. 3, l. 79. It does not seem correct to refer to the optic tectum as a circuit. It is enough to state that it is "an important sensimotor integrator" and drop the word circuit.

p. 4, l. 92, change "like" to "such as"

p. 4, l. 93, delete "very" - contribution is either specific or not

p. 4, l. 94, "successful release of capture swims" is poorly explained jargon. What does release of a capture swim mean? "Capture swim release" jargon is used throughout the manuscript but never clearly defined. Why is the term "release" used? Release from what? Explain precisely for non-specialists.

p. 4, l. 95, These 2 sentences say the same thing: "These data allow us to propose a model in which ITNs are part of a neural circuit that integrates binocular visual information to localize prey and initiate capture swims. To our knowledge, this study is the first description of a neural circuit that can explain binocular information integration needed for prey capture behavior despite the absence of ipsilateral retinotectal projections."

Authors can remove "these data allow us to propose a model" and "to our knowledge, this study is the first description" and combine into 1 strong sentence such as:

"We propose that the ITNs are part of a neural circuit that integrates binocular visual information to localize and capture prey in the absence of ipsilateral retinotectal projections."

p. 5, l. 135 (and in Fig. 3 legend). PVN abbreviation is not defined anywhere in the text.

Discussion

p. 10, l. 257, delete "to the best of our knowledge, first evidence" it is not all that useful

p. 11 l. 303, let = led?

p. 11, l. 307, our data suggest (not suggests)

p. 12, l. 337, change to "which consists of"

Methods

As noted above, authors should use accepted nomenclature and allele designations for all transgenic lines used in this study. In one place (p. 17, l. 426) Tg(elav13:GCaMP5G) is used and in another (l. 486) Tg(HuC:GCaMP5G). Are these different transgenic lines? Allele designations are necessary.

If approximately 100 paramecia were added in the feeding assay, how could the authors get an accurate reading of the number consumed by counting remaining numbers? I am assuming that each separate larva had a precise number initially added. This language should be made clearer. The methods could use some editing (i.e., shorter and clearer) and grammar corrected throughout (e.g., lines 585, and 590 "were (not was) determined").

Figures

The schematic of the larval head is not needed in Fig. 1c'. It doesn't add anything to what is shown by the same drawing in Fig. 1a. At this point in the paper, in its current organization, it is premature to draw lines crossing the tectal midline.

Fig. 2. It is not clear why a GCaMP3 transgenic line was used to trace axonal projections instead of a membrane-tagged fluorescent protein. Nor is it obvious why some experiments utilize GCaMP3 and others GCaMP5.

Fig. 5. Images to validate ITN ablation should be shown in the main Figure (perhaps images for all 12 larvae could be provided in the Supplemental Figure). Also, since authors state elsewhere that in some larvae only one ITN nucleus is "strongly labelled possibly due to genetic variegation" of

the transgenic line they are using, one wonders how this variability affects their ability to assess whether ablations are successful or not?

It is unnecessary to show the "hunting sequence classification" flow chart twice. Remove from Supplemental Fig. 5a.

It is surprising that the authors do not include some supplemental video footage to show the distinct behavioral difference between ITN ablated larvae and unablated controls.

Reviewer #2 (Remarks to the Author):

This is an interesting manuscript. The mechanisms in how binocular visual input is processed in zebrafish with only completely crossed RGC axon projections to the tectum has not been described before. The data therefore represent novel findings of general interest.

Overall, I think the experiments are generally well designed and carried out. However, some points need clarification:

Cell numbers:

The authors present data to show that there are about 30 neurons labelled in the ITN-Gal4 line. However, they do not indicate if this is – judged on the functional imaging results- would represent the entire population of intertectal neurons or only a subset. Given that the functional imaging data was analysed on a voxel-based method, one should repeat these experiments using a nuclear GCaMP line to count the actual cell numbers that are active on the ipsilateral side upon visual stimulation.

Cell arrangement:

In Figure 2B it looks like there is topography in the arrangement of the cell bodies and their projections. Is this true? In addition, is there a link of such topography regarding overall visual space, i.e. where the small dot is seen and being represented in the tectum? This could be discussed.

Labelling of pre- and postsynaptic structures

Figure 3B and C are not clear. This is partially based on the fact that two independent Gal4 drivers were used in simultaneous combination with GCaMP and GFP. It would be helpful for example to show the Brn3c:Gal4, UAS:Syp-GFP alone (without the ITN:Gal4, UAS:GCaMP3) to have better judgement of the basic signal. Similarly, from the images shown, it is difficult to distinguish the GCaMP signal from the psd95-GFP signal (plus the additional complication of expression in some PVNs.).

Cell ablation:

The authors indicate that 12-26 ITNs were ablated in the different fish. Is there a correlation on the number (or location) of cells ablated and the magnitude of effect seen in prey capture? Knowing the total number of cells involved (see previous point) would also give a better judgement of the relative loss of 12 or 26 cells.

In Supplementary Figure 4a it is not absolutely clear if the circle shows the same area in pre- and post-ablated animals. Using some of the other cells as a reference, it seems that the entire image on the post ablation is a bit shifted lower and does therefore not show the entire group of cells seen also in the pre-ablated animal. The authors should make sure the exact same area is shown.

Minor comments:

Line 39: the population is not “novel”, but rather “newly identified”

The authors describe in the abstract that the ITNs respond to prey-like structure. This is true, but they also do respond to other moving visual stimuli, like the ones shown in the initial functional imaging experiments.

Fig 2D: “ventral” is missing at the -272um end of the scale.

Line 385: “Tg(Brn3c:Gal4, UAS:Syp)” should be “Tg(Brn3c:Gal4, UAS:Syp-GFP)”

Reviewer #1 (Remarks to the Author):

The manuscript by Gebhardt et al. describes the characterization of a novel group of cells termed intertectal interneurons (ITNs), which mediate communication between the left and right tectal hemispheres of the larval zebrafish brain. Through calcium imaging and ablation experiments, the authors show that ITNs are motion responsive and important for the visualization and capture of prey. This is an inherently interesting study because it answers a long-standing question of how the fish visual system can functional effectively to integrate binocular cues, when comprised of only contralateral and no ipsilateral retinal projections.

Despite my enthusiasm for this work, I find the organization of ideas somewhat awkward and the writing not always accessible. Use of poorly explained jargon and stacked modifiers do not help if this study is to be understood by a broad readership. These concerns can be easily rectified.

Our response:

Generally, reviewer 1 is enthusiastic about our work saying that is **[...] an inherently interesting study because it answers a long-standing question of how the fish visual system can functional effectively to integrate binocular cues.**

The reviewer is raising concerns that **[...] the organization of ideas [is] somewhat awkward and the writing not always accessible.** However, according to the reviewer these **[...] concerns can be easily rectified.**

We thank the reviewer for the supportive comments. In this revised version we have streamlined the text and we hope it is now an easier read. Below we address specific comments:

Reviewer #1 – Comment #1:

As presented, the premise behind the study seems somewhat artificial – that the authors were examining ipsilateral activation of the tectum by visual stimuli, “predicted the existence of motion responding commissural neurons that connect the tectal hemispheres” and just happened to have a transgenic gene trap line in hand that labels the relevant, bilaterally projecting neurons. It would seem the other way around - that the authors recovered a zebrafish line with an unexpected axonal labeling pattern and explored what the function of the bilaterally projecting neurons might be. This seems far more plausible and, if accurate, should be presented in this manner.

Our response to Comment #1:

It is true that we were aware of the general existence of intertectal axonal connections at the beginning of the project but the function of these axon tracts and

whether they even serve to interconnect tectal neurons was unknown. The experimental observation of visually-evoked activity in the ipsilateral neuropil (**shown in Fig. 1**) preceded our discovery of the *^{ITN}Gal4*-line which in turn provided us with a genetic tool to explore intertectal information transfer.

Reviewer #1 – Comment #2:

The authors are strongly advised to use standardized nomenclature (<https://wiki.zfin.org/display/general/ZFIN+Zebrafish+Nomenclature+Conventions#ZFINZebrafishNomenclatureConventions-4.3>) for labeling their gene trap line. ITNGal4 is not correct according to accepted naming practice in the field. Moreover, seeing as how their new line labels a unique and poorly studied neuronal population, it is of interest to know the genetic locus where the transgenic insertion resides. This could be readily determined by inverse PCR.

Our response to Comment #2:

We now introduce the transgenic insertion as *Gal4ic3034Tg* in the revised manuscript and thereafter refer to it as *^{ITN}Gal4* for ease of reading.

The line was generated serendipitously by injecting an old plasmid derived from the Chien's lab containing the regulatory region of the *Isle2b* gene and a Gal4 reporter. A map of the plasmid is not available nor its full sequence. We tried inverse PCR as we routinely do for the Tol2 transgenic line we generated using the appropriate primers but it did not yield any results. Therefore, we tried whole-genome sequencing but did not receive high enough coverage in the region of the Gal4 reporter to identify the locus of insertion possibly due to DNA degradation. The expression pattern of this is most likely generated by an interaction of the *Islet2b* promoter with the local region of the transgenic insertion. We believe that to know the genetic locus where the transgenic insertion resides will not be informative at this point and although we will continue to investigate it with another full genome sequencing attempt this information is not available at the moment. We are also planning single cell sequencing experiments on ITNs but this work is now beyond the scope of this manuscript. The line has been stable for over 7 years and it will be a resource freely distributed to the community.

Reviewer #1 – Comment #3:

Why were the ITN ablation experiments only performed unilaterally? Is there a difference in the behavioral responses if all of the ITNs are ablated?

Our response to Comment #3:

The hypothesis that ITNs mediate a binocular computation predicts that unilateral loss of ITNs should suffice to produce a behavioural phenotype. We therefore started with unilateral ablations and indeed observed a defect in capture swim release.

Notably unilateral ablation provides a level of internal control for unintended tissue damage that might produce lateralization effects. The absence of such effects, as shown by the indistinguishable turning angle distributions of swim bouts in control and ablated larvae (**Supplementary Fig. 4g**), supports the specificity of the loss-of-function phenotype. In the course of doing these ablations we found that the ITNs are located in close proximity to a network of brain vasculature and that occasionally blood vessel damage resulted in immediate death of the larva. This in turn made it very difficult to perform complete bilateral ablations. In any case, more conservative unilateral ablations proved sufficient to demonstrate a requirement for left and right hemisphere ITNs for generation of capture swims, compatible with our model.

Reviewer #1 – Comment #4:

The model depicted in Fig. 6 does not take into account that ITNs were found to be GABAergic and propose a plausible inhibitory role for these cells. Even in the Discussion, it is confusing that the authors argue that execution of prey capture behavior requires “an ITN-mediated disinhibition in both tectal hemispheres” (does this mean disinhibition of ITNs or ITNs disinhibiting other neurons?). One would think that activation of ITN GABAergic neurons would have an enhanced inhibitory effect. I don’t follow the logic of p. 11, para. 2 to p. 12, para. 1, as presented.

Our response to Comment #4:

The model we propose is very important for the papers message and thus should be explained in a clear and concise manner. We thus revised the mentioned paragraphs (**I302-324**) and the accompanying Figure legend (**I.980-991**) to contain more explanations of how we think inhibitory ITNs could exert their function on capture swim initiation.

Moreover, since tools for optogenetic activation or synaptic inhibition of Gal4 expressing neurons are available, it would be beneficial to examine the behavioral consequences of modulating the activity of ITNs. Such experiments would greatly strengthen this study and help in framing the model.

While *Gal4*-expression in our transgenic line in the mesencephalic tegmentum is sparse and confined to both ITN nuclei, *Gal4* is also expressed in the pineal gland, tectal S1Ns and PVNs (**Fig. 2a**) and quite prominently in the spinal cord. This is important information and has been added to the revised version of the paper (**I.142ff & I. 862f**) Unfortunately, this also suggests that the ^{ITN}*Gal4* line is unsuitable for specific, targeted optogenetic manipulation.

Notwithstanding concerns about specificity of labelling, we have now performed experiments using the ^{ITN}*Gal4*-line to express *UAS:BoTxBLC-GFP* (**Sternberg et al. Curr Bio 2016**, the line was a gift from Claire Wyart, ICM Paris) in order to synaptically silence ITNs (**I.213ff**). We found that ^{ITN}*Gal4*, *UAS: BoTxBLC-GFP* larvae showed a severe reduction in prey consumption (**Supplemental Fig. 4a**) in a prey capture performance assay (**Gahtan et al. 2005**) compared to control siblings. But further analysis of individual larvae revealed that *BoTxBLC-GFP*-expressing larvae did not inflate their swim bladders and thus spent significantly less time swimming as well as covering less distance during prey capture trials (**Supplemental Fig. 4b**). This is unsurprising given the extensive spinal cord expression driven by the transgene. Thus, targeted laser ablations provide a more specific way to examine the consequences of ITN loss-of-function.

Overall, despite labelling other neurons, the ^{ITN}*Gal4*- line proved to be an excellent tool for examining ITN morphology and activity and for guiding targeted laser-ablation of ITNs.

Reviewer #1 – Comment #5:

A few sentences at the start of the Discussion reviewing the differences between vertebrate species in the extent and retention of ipsilateral retinal projections (and summarizing current models to account for this diversity) would be very helpful for a reader to gauge the broader significance of this study. Although there is some mention of the potential existence of similar neurons in other species, it would strengthen this study if the authors could make a better case for conservation of ITN-like neuronal function.

Our response to Comment #5:

We added a paragraph at the beginning of the discussion (**I.270**) reviewing what is known about the extent and retention of ipsilateral retinal projections (IRPs) in vertebrates. In short, it seems that IRPs are an ancestral trait and, consistent with our

study, there does not seem to be a high correlation between their existence and an animal's ability to react to binocular cues. By describing a commissural neural circuit connecting the two tectal halves we offer a possible explanation why some animals without IRPs could still interpret binocular cues.

Reviewer #1 - Minor suggested changes:

Abstract p. 2, l. 39, modify to "a novel commissural neuronal population termed intertectal interneurons (ITNs) that connect the tectal hemispheres"

This has been changed (**see I.37-39**).

Introduction

p. 3, l. 53. Need a better transition sentence between paragraphs 1 and 2 to clearly state that fish have laterally positioned eyes. The issue is not clearly explained here- too cryptic for a general readership as written.

We added a sentence explaining that zebrafish have laterally positioned eyes (**I.56**).

p. 3, l. 79. It does not seem correct to refer to the optic tectum as a circuit. It is enough to state that it is "an important sensimotor integrator" and drop the word circuit.

This has been changed (**I.79**).

p. 4, l. 92, change "like" to "such as"

This has been changed (**I. 92**).

p. 4, l. 93, delete "very"- contribution is either specific or not

This has been changed (**I.93**).

p. 4, l. 94, "successful release of capture swims" is poorly explained jargon. What does release of a capture swim mean? "Capture swim release" jargon is used throughout the manuscript but never clearly defined. Why is the term "release" used? Release from what? Explain precisely for non-specialists.

We agree that this might seem confusing without explanation. We initially aimed to use the neuroethological concept of a “release of a behavior” in the same sense as it was introduced by Konrad Lorenz (**Lorenz, K. & Leyhausen, P. Motivation of Human and Animal Behavior (Van Nostrand Reinhold Company, 1973)**) such that the intensity of internal drive states (e.g. hunger) in concert with external 'releasing' stimuli (e.g. prey in binocular strike zone), would lead to a specific behavioral action (e.g. performing capture strikes). For a more intuitive description, we exchanged “release of a behavior” with “initiation of a behavior” throughout the manuscript (**see e.g. I.93**).

p. 4, l. 95, These 2 sentences say the same thing: “These data allow us to propose a model in which ITNs are part of a neural circuit that integrates binocular visual information to localize prey and initiate capture swims. To our knowledge, this study is the first description of a neural circuit that can explain binocular information integration needed for prey capture behavior despite the absence of ipsilateral retinotectal projections.” Authors can remove “these data allow us to propose a model” and “to our knowledge, this study is the first description” and combine into 1 strong sentence such as: “We propose that the ITNs are part of a neural circuit that integrates binocular visual information to localize and capture prey in the absence of ipsilateral retinotectal projections.”

This has been changed (**I.94-97**).

p. 5, l. 135 (and in Fig. 3 legend). PVN abbreviation is not defined anywhere in the text.

This has been added to where PVNs are first mentioned in the text (**I.123 and I.864**).

All minor comments can be easily addressed.

Discussion

p. 10, l. 257, delete “to the best of our knowledge, first evidence” it is not all that useful

p. 11 l. 303, let = led?

p. 11, l. 307, our data suggest (not suggests)

p.1 2, l. 337, change to “which consists of”

This has been changed.

Methods

As noted above, authors should use accepted nomenclature and allele designations for all transgenic lines used in this study. In one place (p. 17, l. 426) Tg(elavl3:GCaMP5G) is used and in another (l. 486) Tg(HuC:GCaMP5G). Are these different transgenic lines? Allele designations are necessary.

Tg(elavl3:GCaMP5G) and HuC:GCaMP5G are different descriptors for the same transgenic line. We refer to them consistently now according to standardized zfin nomenclature in the revised manuscript.

If approximately 100 paramecia were added in the feeding assay, how could the authors get an accurate reading of the number consumed by counting remaining numbers? I am assuming that each separate larva had a precise number initially added. This language should be made clearer.

We aimed for 100 paramecia per larvae per experiment but due to inevitable fluctuations when applying the paramecium suspension to the dish, the precise paramecia number added per dish had to be precisely determined retrospectively from video tracking data. An explanation was added to the revised manuscript **(I.554)**.

The methods could use some editing (i.e., shorter and clearer) and grammar corrected throughout (e.g., lines 585, and 590 "were (not was) determined").

We revised the methods section and corrected the grammatical error pointed out **(I.585 and I.590)**.

Figures

The schematic of the larval head is not needed in Fig. 1 1c'. It doesn't add anything to what is shown by the same drawing in Fig. 1a. At this point in the paper, in its current organization, it is premature to draw lines crossing the tectal midline.

The dotted lines in Figure 1c' were not intended to indicate potential interhemispheric connections but rather the curved ROI along whose longer axis the average of relative density (active pixels) was calculated. We modified the figure and the legend to clarify that.

Fig. 2. It is not clear why a GCaMP3 transgenic line was used to trace axonal projections instead of a membrane-tagged fluorescent protein. Nor is it obvious why some experiments utilize GCaMP3 and others GCaMP5.

Although GCaMP3 was in the genetic background for the neurite tracing experiments, the GCaMP3-signal in single neurites was too weak to be used for the tracing of individual ITN projections. Therefore, we instead injected a *UAS:GFP*, *UAS:tagRFPCaax* plasmid into the ^{ITN}Gal4 zygotes to transiently label a small number of ITNs. In our hands this construct randomly labeled ITNs in red (membrane-targeted), green (not membrane targeted) or both thus facilitating ITN identification and tracing. GCaMP3 and GCaMP5 were used due to the availability of the respective lines. We do not consider this to be a significant issue because we did not compute or compare inferred firing rates between the GCaMP variants.

Fig. 5. Images to validate ITN ablation should be shown in the main Figure (perhaps images for all 12 larvae could be provided in the Supplemental Figure). Also, since authors state elsewhere that in some larvae only one ITN nucleus is “strongly labelled possibly due to genetic variegation” of the transgenic line they are using, one wonders how this variability affects their ability to assess whether ablations are successful or not?

ITN pre- and post-ablation panels from **Supplemental Figure S4a** have been moved to the main **Figure 5a**.

Additionally, a statement has been added to the Materials and Methods explaining that due to genetic variegation of the ^{ITN}Gal4-line, we selected larvae for ablation in which at least 12 ITNs (on one side) were labelled (**I. 523**). This represents ~50% of the estimated population (see also comments to Reviewer 2 below).

It is unnecessary to show the “hunting sequence classification” flow chart twice. Remove from Supplemental Fig. 5a.

This has been changed.

It is surprising that the authors do not include some supplemental video footage to show the distinct behavioral difference between ITN ablated larvae and unablated controls.

We added two movies showing a 6dpf control- or ITN-ablated larvae respectively while they were hunting paramecia. In both cases prey was localized in front of the larvae in a position that leads to high probability of capture swim initiation in control larvae (**Supplementary Movie 1**) but does not in ITN-ablated larva (**Supplementary Movie 2**).

Reviewer #2 (Remarks to the Author):

This is an interesting manuscript. The mechanisms in how binocular visual input is processed in zebrafish with only completely crossed RGC axon projections to the tectum has not been described before. The data therefore represent novel findings of general interest. Overall, I think the experiments are generally well designed and carried out. However, some points need clarification:

Our response:

Reviewer 2 also considers our study to be **of general interest** and **the experiments [being] generally well designed and carried out**. Some points however **need clarification**. We thank the reviewer for giving us the opportunity to revise and improve the manuscript and we would like to respond to the individual points in the following:

Reviewer #2 – Comment #1:

Cell numbers:

The authors present data to show that there are about 30 neurons labelled in the ITN-Gal4 line. However, they do not indicate if this is – judged on the functional imaging results- would represent the entire population of intertectal neurons or only a subset.

Given that the functional imaging data was analysed on a voxel-based method, one should repeat these experiments using a nuclear GCaMP line to count the actual cell numbers that are active on the ipsilateral side upon visual stimulation.

Our response to Comment #1:

We initially estimated the number of ITNs to be around 30 on either side of each animal based on the maximum labelling that we saw in ^{ITN}*Gal4* larvae. To further estimate ITN cell numbers, we have now imaged 18 randomly selected larvae at 6 dpf and registered them to the zbrain tErk reference brain (**Randlett et al. 2015**) using the Computational Morphometry Toolkit (CMTK, **Rohlfing and Maurer, 2003**). Then we counted the number of ITN cell bodies within the two regions identified as ITN nuclei in an average intensity larva (**Figure for the reviewer, below**).

The average number of ITN cell bodies in both nuclei was 32 in total, with the minimum being 22 (12 on the left, 10 on the right) and the maximum being 48 (27 on the left, 21 on the right). We do not have evidence to think that this variation reflects actual differences in ITN numbers per individual but rather reflects variegated ITN labelling in the *ITN Gal4* line.

For the second part of the comment, we assume that by “**actual cell numbers that are active on the ipsilateral side upon visual stimulation**” the reviewer is referring to PVN activation ipsilateral to the stimulated eye. However, we did not see activation

of ipsilateral PVN cell bodies (**Figure 1c and c'**) and for this reason used a voxelwise analysis to evaluate visually evoked responses in the tectal neuropil.

If the reviewer is referring to ITN activity, we note that only ITNs contralateral to the visual stimulus responded to prey-like stimuli **Figure 4c**. The responses of these cells were indeed assessed by assigning ROIs to single neurons (i.e. the data in **Figure 4** is cellwise rather than voxelwise). Due to variegation of GCaMP expression, we could not estimate the fraction of the responding ITN population in these experiments, and the activity raster represents neurons sampled across multiple animals (N=5).

Reviewer #2 – Comment #2:

Cell arrangement:

In Figure 2B it looks like there is topography in the arrangement of the cell bodies and their projections. Is this true? In addition, is there a link of such topography regarding overall visual space, i.e. where the small dot is seen and being represented in the tectum? This could be discussed.

Our response to Comment #2:

It seems that there is a relationship between ITN cell body position and where the neurite extends through the tectal neuropil and where it crosses the midline. We would be hesitant though to claim the existence of an actual topographic map because we unfortunately do not have a sufficient number of single labeled ITNs to make that claim.

Reviewer #2 – Comment #3:

Labelling of pre-and postsynaptic structures

Figure 3B and C are not clear. This partially based on the fact that two independent Gal4 drivers were used in simultaneous combination with GCaMP and GFP. It would be helpful for example to show the Brn3c:Gal4, UAS:Syp-GFP alone (without the ITN:Gal4, UAS:GCaMP3) to have better judgement of the basic signal. Similarly, from the images shown, it is difficult to distinguish the GCaMP signal from the psd95-GFP signal (plus the additional complication of expression in some PVNs.)

Our response to Comment #3:

Thanks to the Reviewers question we realized that we made a minor mistake in the description of **Figure 3B**.

Figure 3B was created by imaging larvae of a cross of a *Tg(Brn3c:Gal4, UAS:Syp-GFP)* (**Figure for the reviewer, above**) and the *^{ITN}Gal4* line. During the pre-screening for imaging, larvae containing the *Brn3c:Gal4* transgene were actually removed and only animals were used that contained the *^{ITN}Gal4* insertion and *UAS:Syp-GFP*. In addition to ITNs, the *^{ITN}Gal4* transgene also labels SINs, a few PVNs, the pineal gland and the spinal cord. In the description for the original **Figure 3B** we mistook synapses in the SO and SFGS to be RGC inputs originating from the *Brn3c-Gal4* transgene while they are indeed synapses of labelled SINs in the upper layers. This does not alter our conclusions, in fact it simplifies the interpretation of this figure. This has been clarified in a revised **Figure 3B** and the Methods (**I. 381**). In order to improve visibility of the labelling of postsynaptic specializations of ITNs and making it easier to distinguish from PVN labeling we traced the ITN and color-coded its trajectory according to its z-position in a 3D stack. This was added to **Figure 3c** as an inset next to the raw data.

Reviewer #2 – Comment #4:

Cell ablation:

The authors indicate that 12-26 ITNs were ablated in the different fish. Is there a correlation on the number (or location) of cells ablated and the magnitude of effect seen in prey capture? Knowing the total number of cells involved (see previous point) would also give a better judgement of the relative loss of 12 or 26 cells.

Our response to Comment #4:

In order to answer if there is a correlation between the number of ITNs ablated and the magnitude of the effect seen in prey consumption we performed 23 ITN ablation experiments with ablated cell numbers lower than 12 ITNs to cover the entire range of ITN numbers seen in *^{ITN}Gal4* larvae. Subsequently we performed prey consumption assays as previously described. Plotting the number of ablated ITNs vs. the feeding coefficients (coefficient A of an exponential function $f(x) = \exp(A \cdot x)$ fitted to the individual feeding curves) for each larva revealed a strong trend such that the more ITNs were ablated the more severe was the deficit in prey consumption (**Supplementary Figure 4c**).

In Supplementary Figure 4a it is not absolutely clear if the circle shows the same area in pre-and post-ablated animals. Using some of the other cells as a reference, it seems that the entire image on the post ablation is a bit shifted lower and does therefore not show the entire group of cells seen also in the pre-ablated animal. The authors should make sure the exact same area is shown.

This indeed escaped our notice and was changed in this revised version of the paper. As per suggestion of Reviewer 1 we moved these panels to main **Figure 5a**.

Reviewer #2 – Minor comments:

Line 39: the population is not “novel”, but rather “newly identified”

This has been changed in the revised version of the manuscript (**p.2, l.37f**).

The authors describe in the abstract that the ITNs respond to prey-like structure. This is true, but they also do respond to other moving visual stimuli, like the ones shown in the initial functional imaging experiments.

That is true. We attenuated this statement in the revised version of the manuscript (**p.2, l.40**).

Fig 2D: “ventral” is missing at the -272um end of the scale.

This has been changed in the revised version of the figure.

Line 385: “Tg(Brn3c:Gal4, UAS:Syp)” should be “Tg(Brn3c:Gal4, UAS:Syp-GFP)”

This has been changed in the revised version of the manuscript (**l.381**).

REVIEWERS' COMMENTS:

Reviewer #1 (Remarks to the Author):

The authors addressed many of the points raised in the prior reviews and made necessary corrections. However, it is disappointing that they failed to resolve the genetic basis of the Gal4 transgenic line that labels the intertectal neurons.

Overall, I think this is an interesting study that provides new information about processing of visual information for successful prey capture.

There are a few additional changes that would improve the manuscript:

The paragraph from lines 71 to 78 seems out of place in the Introduction. I suggest moving it to follow line 50.

Overall, the Methods section could benefit from additional editing.

Throughout the Methods, the word "then" is overused. For example, in the section entitled "Image analysis for Calcium imaging experiments" (l. 476), "then" could be removed from nearly every sentence.

- l. 478, unclear why "motion corrected" is hyphenated
- l. 555 "since the larvae received food more than 12 hours ago" - change ago to earlier
- l. 563, "number of living preys per dish" – should be prey not preys
- l. 570 "Each recording lasted approximately between 5 to 10 minutes" – remove approximately
- l. 572 "For the analysis of ITN-ablated larvae" – unclear why this text is underlined
- l. 629 "success or failure was (not were) recorded"
- l. 633, Data analysis
- l. 822, with input (not inputs)

Reviewer #2 (Remarks to the Author):

In this revised version of the manuscript, the authors have in my opinion appropriately addressed my concerns raised previously. I am therefore satisfied with their response. The inclusion of additional data and the clarification of the text/figures has significantly improved the manuscript overall.

Reviewer #1 (Remarks to the Author):

The authors addressed many of the points raised in the prior reviews and made necessary corrections. However, it is disappointing that they failed to resolve the genetic basis of the Gal4 transgenic line that labels the intertectal neurons.

Overall, I think this is an interesting study that provides new information about processing of visual information for successful prey capture.

There are a few additional changes that would improve the manuscript:

Our response:

We thank the reviewer for the helpful comments. Below we address the remaining comments:

The paragraph from lines 71 to 78 seems out of place in the Introduction. I suggest moving it to follow line 50.

Overall, the Methods section could benefit from additional editing.

Throughout the Methods, the word “then” is overused. For example, in the section entitled “Image analysis for Calcium imaging experiments” (l. 476), “then” could be removed from nearly every sentence.

l. 478, unclear why “motion corrected” is hyphenated

l. 555 “since the larvae received food more than 12 hours ago” - change ago to earlier

l. 563, “number of living preys per dish” – should be prey not preys

l. 570 “Each recording lasted approximately between 5 to 10 minutes” – remove approximately

l. 572 “For the analysis of ITN-ablated larvae” – unclear why this text is underlined

l. 629 “success or failure was (not were) recorded”

l. 633, Data analysis

l. 822, with input (not inputs)

Our response:

All points have been corrected in the revised manuscript.

Reviewer #2 (Remarks to the Author):

In this revised version of the manuscript, the authors have in my opinion appropriately addressed my concerns raised previously. I am therefore satisfied with their response. The inclusion of additional data and the clarification of the text/figures has significantly improved the manuscript overall.

Our response:

We thank the reviewer for the comments that helped to improve the manuscript tremendously.